# Aptamers as Potential Therapeutic Tools for Ovarian Cancer: Advancements and Challenges

**DOI:** 10.3390/cancers15215300

**Published:** 2023-11-06

**Authors:** Wojciech Szymanowski, Anna Szymanowska, Anna Bielawska, Gabriel Lopez-Berestein, Cristian Rodriguez-Aguayo, Paola Amero

**Affiliations:** 1Department of Biotechnology, Medical University of Bialystok, 15-222 Bialystok, Poland; wojtekszymanowski@wp.pl (W.S.); anna.bielawska@umb.edu.pl (A.B.); 2Department of Experimental Therapeutics, The University of Texas MD Anderson Cancer Center, Houston, TX 77030, USA; szymanowska_anna@wp.pl (A.S.); glopez@mdanderson.org (G.L.-B.); crodriguez2@mdanderson.org (C.R.-A.); 3Center for RNA Interference and Non-Coding RNA, The University of Texas MD Anderson Cancer Center, Houston, TX 77030, USA; 4Department of Cancer Biology, The University of Texas MD Anderson Cancer Center, Houston, TX 77030, USA

**Keywords:** aptamers, ovarian cancer, tumor microenvironment, targeted delivery system, therapeutics

## Abstract

**Simple Summary:**

Aptamers are versatile molecules for cancer treatment. They can be used to target specific molecules, conjugated to additional nucleic acid moieties or payloads. In this review, we will summarize the advancements and challenges related to the use of aptamers for the treatment of ovarian cancer. We will specifically focus on aptamers as therapeutics alone or conjugated to non-coding RNA, nanoparticles, and multivalent aptamers, highlighting their potential as therapeutics by providing an overview of the latest advancements in the field reported in the literature and discussing the main challenges that need to be overcome to improve the effectiveness of aptamers for the treatment of ovarian cancer.

**Abstract:**

Ovarian cancer (OC) is the most common lethal gynecologic cause of death in women worldwide, with a high mortality rate and increasing incidence. Despite advancements in the treatment, most OC patients still die from their disease due to late-stage diagnosis, the lack of effective diagnostic methods, and relapses. Aptamers, synthetic, short single-stranded oligonucleotides, have emerged as promising anticancer therapeutics. Their ability to selectively bind to target molecules, including cancer-related proteins and receptors, has revolutionized drug discovery and biomarker identification. Aptamers offer unique insights into the molecular pathways involved in cancer development and progression. Moreover, they show immense potential as drug delivery systems, enabling targeted delivery of therapeutic agents to cancer cells while minimizing off-target effects and reducing systemic toxicity. In the context of OC, the integration of aptamers with non-coding RNAs (ncRNAs) presents an opportunity for precise and efficient gene targeting. Additionally, the conjugation of aptamers with nanoparticles allows for accurate and targeted delivery of ncRNAs to specific cells, tissues, or organs. In this review, we will summarize the potential use and challenges associated with the use of aptamers alone or aptamer–ncRNA conjugates, nanoparticles, and multivalent aptamer-based therapeutics for the treatment of OC.

## 1. Introduction

Ovarian cancer (OC) is the most lethal gynecologic malignancy leading to death among women worldwide. The incidence and mortality rates of OC have been steadily increasing each year. In 2020, almost 314,000 new cases and 207,000 deaths have been reported [1,2]. Although the survival rates have improved during the past four decades, 70% of OC patients still succumb to the disease within ten years after diagnosis [3,4]. Current treatments are debulking surgery, followed by platinum-based chemotherapy; however, the overall clinical outcomes for OC patients continue to be unsatisfactory. This is primarily due to the occurrence of relapse and extensive metastases (in stages III and IV) at the time of diagnosis for most OC patients [5,6].

Recent discoveries have highlighted the significant role that the tumor microenvironment (TME) plays in cancer development and resistance to chemotherapy. Within the TME, various cell types are identified, including immune cells, endothelial cells, and cancer-associated fibroblasts (CAFs). These cells contribute not only to cancer progression but also to evading immune surveillance [6,7,8]. The heterogeneity of the TME in OC was demonstrated in 2022 by Stur et al. Using spatial RNA sequencing, Stur et al. conducted a transcriptomic analysis of tumor tissues from individuals responding differently to carboplatin and paclitaxel treatments. They found that the TME in patients with poor response to therapy had a lower infiltration of immune cells, while those who responded excellently exhibited a high percentage of these cells. This observation underscores the critical role of immune cells in responding to chemotherapy. The communication between TME cells relies partially on cytokines, proteins, and non-coding RNAs (nc-RNAs) transported by exosomes. Therefore, understanding the interaction between these cells and the TME’s composition may help in designing novel targeted therapies using dCas9 systems and aptamers [9,10].

Aptamers are a promising class of anticancer therapeutics that are capturing the attention of researchers and clinicians due to their versatility [11]. Their main characteristic is their ability to recognize molecule targets based on their three-dimensional (3D) structures [12]. They have proven to be invaluable tools in the field of drug discovery, enabling the identification and characterization of novel therapeutic targets [12,13]. Aptamers can be utilized to explore the molecular pathways involved in cancer development and progression, shedding light on key biomarkers and signaling mechanisms [14]. Furthermore, aptamers exhibit immense potential in the realm of drug delivery systems by conjugating aptamers with other therapeutic agents, such as chemotherapeutic drugs or nanoparticles (NPs) [15]. This approach minimizes off-target effects, enhances drug efficacy, and reduces systemic toxicity, ultimately leading to improved patient outcomes [16].

The distinctive genetic hallmarks of OC and the well-established oncogenes present a favorable opportunity for the implementation of gene-based therapies like RNA interference (RNAi) [17,18]. However, the practical application of non-coding RNAs (ncRNAs) encounters challenges associated with their instability in the bloodstream and potential toxicity within living organisms due to off-target effects [18,19]. In response to these limitations, a promising approach has emerged involving the integration of aptamers, which possess remarkable targeting specificity, with ncRNA technology [20]. This strategic fusion holds immense potential for achieving precise and efficient gene targeting. Furthermore, there is a growing interest in utilizing NPs that are conjugated with aptamers as delivery systems for ncRNAs in the treatment of OC [21,22]. This modification enables the targeted delivery of ncRNAs to specific cells, tissues, or organs, facilitating their internalization with a high degree of accuracy [23].

In this review, we aim to summarize the present status and challenges associated with the use of aptamers for the diagnosis and treatment of OC. We focused on aptamers as therapeutics alone or conjugated to non-coding RNAs (ncRNAs) and NPs, highlighting their potential as theragnostic agents. By examining the existing literature and research advancements, we aim to shed light on the current landscape and identify the key challenges that need to be addressed to enhance the efficacy of aptamer-based approaches in OC treatment.

## 2. Role of Tumor Microenvironment in Ovarian Cancer Progression

OC can be categorized into three primary types: epithelial (the most common type), germ cell, and sex cord-stromal. However, the latter two types, germ cell, and sex cord–stromal, constitute only approximately 5% of all OC cases. Among epithelial ovarian cancers (EOCs), there are four main histologic subtypes: serous, endometrioid, mucinous, and clear cell. Serous tumors can be further classified into high-grade serous carcinomas (HGSC) and low-grade serous carcinomas (LGSC). HGSCs make up around 70% to 80% of all subtypes of EOC, while LGSCs account for less than 5%. The remaining subtypes, namely endometrioid, mucinous, and clear cell, represent approximately 10%, 3%, and 10% of cases, respectively [8,24,25]. EOC is a heterogeneous disease characterized by the presence of tumors displaying diverse histologies, grades, and molecular and microenvironmental characteristics. These factors collectively influence the response to treatment and overall outcomes. As previously mentioned, EOC can be categorized into four distinct subtypes, each characterized by unique patterns of presentation, clinical outcomes, and responses to therapeutic interventions. These variations in behavior are rooted in the intrinsic biology of the tumor, which ultimately impacts prognosis and overall outcome [25,26].

Increased shreds of evidence indicate that non-cancer cells present in the TME play different significant roles in tumorigenesis, metastasis, and chemoresistance. These effects are mediated through various signaling interactions, which can occur through direct cell-to-cell interactions or the release of soluble factors, either independently or encapsulated. Unlike some other solid tumors, EOC often exhibits a non-solid component in its TME—the ascites fluid that frequently accompanies advanced stages of the disease. TME refers to a specific niche, whether primary or metastatic, where tumor cells interact with the surrounding host stroma, which includes immune cells, endothelial cells, fibroblasts, and metabolites. Understanding and targeting the TME has become crucial in developing novel therapeutic strategies. The significance of the TME in the initiation and progression of OC, as well as its role in conferring resistance to anti-cancer treatments, is increasingly recognized. The TME provides an opportunity for the development of promising anti-cancer agents. To this day, several agents gained FDA approval, with others currently undergoing phase II/III trials. These agents include angiogenesis inhibitors, immune-checkpoint inhibitors, and PARP inhibitors [6,7]. The progression and metastasis of OC cells can be facilitated by the interactions among macrophages, T cells, natural killer (NK) cells, fibroblasts, and a variety of chemokines and cytokines [27]. Figure 1 shows a schematic representation of these interactions within the OC TME.

Numerous epidemiological studies have established a correlation between obesity and various cancer types [28,29]. White adipose tissue contains a diverse array of cell types, including adipocytes, adipose stromal cells, and immune cells, collectively contributing to cancer cell proliferation and metastasis. Additionally, adipocytes release adipokines that can drive oncogenic signaling, angiogenesis, and immunomodulation, creating a favorable environment for cancer cell survival and proliferation [30]. In the context of OC, the omentum, a prominent abdominal fat pad, is a common site of metastasis. OC invading the omentum encounters a rich adipose tissue environment, with adipocytes providing metabolic support through the supply of fatty acids [31]. Furthermore, adipose tissue contains stromal cells, particularly adipose stromal cells (ASCs), which have been implicated in promoting resistance to paclitaxel and carboplatin in OC cells [7]. Notably, arachidonic acid secreted by adipocytes plays a role in OC chemoresistance [30]. Addressing the impact of adipose tissue on chemoresistance may be a crucial consideration in anticancer therapy for obese patients [32].

Cancer-associated fibroblasts (CAFs) residing within the TME are key stromal cells known for their role in collagen production within solid tumors. In vivo models have linked CAFs to tumor progression and immune response modulation. They provide redox support to OC cells by supplying glutathione (GSH), reducing cisplatin concentration in the nucleus, and mitigating its cytotoxicity [6,7,33]. CAFs are active contributors to various tumor components and influence essential aspects of solid cancers. They express Dickkopf-3 (DKK3), connecting YAP/TAZ and HSF1 signaling pathways and promoting pro-tumorigenic behaviors [6,34]. CAFs also contribute to chemotherapy resistance by interacting with microvascular endothelial cells (MECs) within the OC microenvironment. CAFs also regulate epithelial cancer cell autophagy by releasing pro-inflammatory cytokines, autophagy-derived substrates, and metabolites, contributing to OC progression [35].

Cancer stem cells (CSCs), also referred to as tumor-initiating or sphere-forming cells, constitute a distinct subset of malignant cells responsible for cancer relapse and chemoresistance, possessing self-renewal, tumorigenicity, and pluripotency. CSCs have been isolated in various solid malignancies across different anatomical sites [36,37]. Targeting CSCs therapeutically faces challenges due to their resemblance to non-cancerous cells, limiting treatment opportunities. The dynamic nature of CSCs transitioning between differentiated cells and CSCs adds complexity. CSC plasticity and heterogeneity, shaped by the TME, further complicate therapeutic strategies [36].

MicroRNAs (miRNAs) play a crucial role in regulating various cellular functions by post-transcriptionally suppressing target genes, impacting cell development, differentiation, metabolism, aging, inflammation, and immune responses. Due to their small size, miRNAs can simultaneously target multiple genes, coordinating post-transcriptional expression programs. In cancer, including OC, miRNAs are implicated in pathogenesis and chemoresistance, making them attractive therapeutic targets. Some miRNAs act as tumor suppressors when downregulated, while others with aberrant expression promote oncogenesis. In OC, miRNAs also play a role in chemoresistance by regulating multidrug resistance (MDR) transport proteins and non-transporter genes [7,38].

Exosomes, carry miRNAs, messenger RNA (mRNA), and proteins. In advanced OC, exosomes, along with other extracellular vesicles, are detected in ascites during initial diagnosis and disease recurrence. These exosomal miRNAs are transferred in a paracrine manner from tumor cells to neighboring cells, impacting metastasis, disease progression, and chemoresistance. Therefore, exosomal miRNAs are being investigated as predictive biomarkers and prognostic biomarkers. Exosomes can also modulate macrophages into tumor-associated macrophages (TAMs), creating a pro-metastatic environment by releasing immunosuppressive factors within the TME. Notably, exosomes from OC patients contain proteins involved in adhesion, angiogenesis, metabolism, and proliferation [7,39,40].

Tumor-associated macrophages (TAMs) are a specialized population of macrophages found within tumor environments, playing pivotal roles in inflammation and tumorigenesis. Extensive research has revealed their significant influence on the dynamic interplay between the TME and cancer cells, with strong associations with key aspects of tumor progression, including invasion and metastasis [6,7,8]. TAMs exhibit versatility, adopting two primary functional phenotypes: M1 and M2. M1 macrophages, stimulated by factors like interferon-gamma (IFN-γ), bacterial lipopolysaccharide (LPS), and granulocyte-macrophage colony-stimulating factor (GM-CSF), have a pro-inflammatory profile, secreting cytokines such as IL-1, IL-12, TNF-α, and CXCL12 (stromal cell-derived factor 1). They display potent cytotoxicity, support tumor suppression, and boost immune responses [6,41]. In contrast, exposure to cytokines like IL-4, IL-10, and IL-13 leads monocytes to differentiate into M2 macrophages. M2 macrophages are anti-inflammatory and promote pro-tumor activity. During the immune evasion phase of tumor development, the TME, driven by factors like IL-4 and IL-13 from cancer cells, fosters an immunosuppressive milieu that encourages monocyte differentiation into M2 macrophages. These M2 macrophages subsequently contribute to tumor growth and progression. In OC, TAMs primarily exhibit an M2 phenotype, closely linked to various malignant characteristics of ovarian tumors, including invasion, angiogenesis, metastasis, and early recurrence. Research has highlighted the prognostic value of TAMs in OC, particularly through the assessment of ratios like M1/M2 and M2/TAMs. However, the overall density of TAMs in ovarian tumors has not demonstrated significant prognostic significance [6,8,27,41,42,43].

Endothelial cells (ECs) are central to the TME, facilitating oxygen and nutrient transport while playing a vital role in angiogenesis, crucial for OC progression and peritoneal spread. However, our understanding of the mechanisms remains limited. ECs are integral to the TME and closely linked to angiogenesis. Key regulators of angiogenesis, including VEGF and angiopoietins, have been associated with unfavorable clinical outcomes, making them potential therapeutic targets. Notably, angiogenesis inhibitors like bevacizumab have shown promising efficacy in phase III trials for OC treatment [6,8,44,45].

Tumor-associated neutrophils (TANs) have a crucial role in responding to inflammation, including that in the TME. The TME is a complex environment where various factors from both tumor and stromal cells induce neutrophil migration and modulate their function for either pro- or anti-cancer effects. While research has explored TANs’ role in promoting or inhibiting tumors, there are gaps in understanding the specific factors that recruit neutrophils to tumors and the functions of TANs. TANs in the TME are adaptable, exhibiting different phenotypes. They can be N1, with anti-tumor activity, or N2, with pro-tumoral effects. TGFβ plays a key role in polarizing TANs, suppressing N1, and promoting N2 phenotypes. However, the functional plasticity of TANs in human OC is not well studied [6,42,46,47].

## 3. Aptamers—Promising Tools for Cancer Treatment

The main factor in the selection of a cancer treatment regimen is based on the place of tumor formation and its stage. Recent research has demonstrated that the molecular genetics of cancer also play a crucial role in the determination of effective treatment [48]. The diagnosis of genetic disorders allows for the identification of genes with an increased or decreased expression level. The use of drugs targeted at specific molecular targets may increase the effectiveness of treatment as well as reduce the risk of side effects associated with traditional chemotherapy [49]. One of the techniques used in targeted therapy is aptamers. Aptamers may be promising anti-cancer therapeutics due to their ease of modification and functionalization [50]. Due to the mechanism of action, aptamers can be divided into three groups: antagonists (blocking protein-protein interactions, e.g., AS1411 blocking nucleolin, AP-50 inhibiting NF-kB) [51], agonists (stimulating receptor activation), and bi-specific aptamers (e.g., MRP1-CD28, which inhibits CTLA-4 and stimulates CD28) [52]. Additionally, aptamers can also be used as carriers of chemotherapeutic agents to deliver them to target tissues (e.g., doxorubicin) [49].

### 3.1. Potential New Target for Aptamer and Aptamer Chimera Development for the Treatment of OC

The potential applications of aptamers in the diagnosis and treatment of tumors, particularly in the context of OC, offer substantial opportunities. Simultaneously, there is a need to identify additional OC biomarkers apart from those currently known and to develop aptamer-based probes tailored for detecting proteins present in low quantities or for facilitating in vivo imaging. These advancements are crucial for enabling early-stage diagnosis of OC [15]. Moreover, given the relatively small molecular size of aptamers, it is imperative to explore suitable materials that can reduce their clearance rate within the body. Combining aptamers with anti-tumor drugs presents an avenue to enhance therapeutic efficacy while mitigating adverse effects. Additionally, the identification of valuable gene targets is essential, and further research should investigate nanomaterials loaded with aptamers, miRNAs, LNAs (locked nucleic acids), and siRNAs to modulate the expression of specific target genes. The ultimate goal is to design aptamer/drug systems that exhibit high specificity for tumor tissues in vivo while minimizing distribution in other tissues. As of now, several aptamers have been documented for their ability to target proteins associated with OC, including aptamers targeting MUC1, CA125, HE4, CD44, and programmed death-ligand 1 (PD-L1) [53,54,55,56,57].

Simaeys et al. have reported the discovery of a set of aptamers capable of specifically recognizing the ovarian clear-cell carcinoma (OCCC) subtype TOV-21 G [58]. Subsequently, they identified stress-induced phosphoprotein 1 (STIP1) as one of the target proteins, indicating its potential utility as a biomarker for OC [59]. The pursuit of identifying novel targets for the development of aptamers and aptamer chimeras as potential therapeutic agents in OC requires a comprehensive understanding of the disease and the intricate molecular pathways involved. To this end, Lee and collaborators have conducted an extensive multi-omics analysis characterized by high-depth scrutiny to discern disparities in the molecular and cellular attributes of highly clinically annotated samples of HGSC originating from both primary and multiple metastatic sites [60]. They observed a notable increase in the loss of copies of the *NF1* gene, as well as reduced levels of its RNA and protein products, fewer cytotoxic T-lymphocyte precursors (CTLPs), and a significantly higher presence of strongly binding neoantigens in the group of patients who underwent primary complete gross resection (R0) compared to those who received neoadjuvant chemotherapy (NACT). Furthermore, the R0 group exhibited a significant rise in T-cell infiltration and a decrease in macrophage numbers. Transcriptomic and proteomic differences were identified in this group [60,61]. Despite differences in copy number variation and T-cell infiltration between the two groups, these disparities were not as extensive as anticipated. These findings suggest that critical molecular distinctions between such tumors cannot be comprehensively unveiled through bulk molecular analyses. Consequently, the researchers conducted an extensive spatial examination of both the epithelial and stromal components within tumor tissue to investigate the heterogeneity of HGSC and explore how the dynamic interactions between the tumor and its microenvironment vary between individuals who respond well and poorly to NACT. To obtain these insights, they conducted an in-depth spatial transcriptomic profiling directly within the tissue [62,63].

In a comprehensive multiomics examination of tumor profiles, Stur et al. compared patients with HGSC who survived for a minimum of 10 years with those who survived for only 3 years. Their investigation identified *TMEM62* as a potential therapeutic target to enhance survival. The overexpression of *TMEM62* was distinctive in the first group of tumors but not observed in the second group, and this increased expression was associated with alterations in cell survival pathways, including upregulation of senescence markers. Furthermore, they observed that the *MAL* gene exhibited the highest upregulation in patients with short-term survival. These findings suggest that the restoration of *TMEM62* could represent a novel approach to the treatment of HGSC. Importantly, these discoveries may hold implications for the development of biomarkers and intervention strategies aimed at improving patient outcomes [63]. Two pivotal discoveries emerged from their investigations: first, they underscored the significance of the stromal component as a potential driver of suboptimal responses to chemotherapy, and second, they identified distinct clusters (in terms of their distribution and composition) in tissues from patients who responded well to NACT compared to those who responded poorly, both at a collective and individual patient level. These clusters could only be discerned through the in situ technique. Notably, when focusing on the epithelial-mesenchymal transition pathway, they observed substantial variations in its upregulation or downregulation among different regions within the same tumor tissue, as well as among specific areas in the tissues across the entire patient cohort. This observation suggests that particular cell populations may contribute to the inadequate response to therapy [9].

In a separate study, Ye et al. conducted an integrative genomic and transcriptomic profiling aimed at identifying molecular subtypes and prognostic markers in OCCC, with a specific focus on immune-related pathways. Through this integrated analysis, they delineated two distinct OCCC molecular subtypes with differing functional characteristics and prognoses. Notably, the immune subset exhibited enrichment in PD-1 and PI3K-AKT-mTOR signaling pathways. Leveraging their data, the researchers constructed a robust prognostic immune signature for OCCC patients, which was not only derived from their dataset but also validated in publicly available repositories. The potential of the immune/non-immune classification as a predictive tool for targeted therapy warrants further investigation through prospective studies [64].

Wang et al. conducted a comprehensive multiomics investigation focused on OC, revealing a notable degree of tumor heterogeneity within this disease. They acquired high-resolution profiles of HGSC, meticulously dissecting both the intertumor and intratumor heterogeneities. This effort led to the identification of numerous potentially crucial variations occurring during tumorigenesis and the elucidation of regulatory networks within HGSC. Through utilizing integrated multiomics analyses, it was revealed that the increased activity of interferon signaling and metallothioneins stemmed from a combination of demethylation in their promoters and hypomethylation in satellite regions and LINE1. Furthermore, the investigation uncovered potential critical transcription factors that govern glycolysis by leveraging chromatin accessibility data. Interestingly, observed patterns in gene expression and DNA methylation remained consistent in both primary and abdominal metastatic tumor cells of matched genetic lineage, suggesting that metastatic cells may potentially exist as subclones within primary tumors. Notably, cancer cell lineages exhibiting heightened residual DNA methylation levels, along with elevated expression of CCN1 and HSP90AA1, displayed a heightened propensity for metastasis. Their study has provided invaluable resources for enhancing our comprehension of the molecular attributes inherent to HGSC. Furthermore, it holds promise for advancing the diagnosis and personalized therapeutic strategies for this disease [65].

Guo et al. integrated single-cell sequencing data from 12 OC patients to construct a comprehensive cell atlas containing normal epithelium, primary carcinoma, and metastatic carcinoma states. Within this atlas, they pinpointed an upregulated gene called *RAB13*, which had not previously been associated with OC. Subsequently, they conducted experiments to validate *RAB13*’s pro-metastatic effects on OC cell lines, demonstrating its promotion of cell migration and invasion in vitro. Further investigations delved into the clinical implications of *RAB13*’s expression levels using datasets from the Cancer Genome Atlas (TCGA), revealing a significant association between RAB13 expression and adverse prognosis and tumor progression. Additionally, the study employed predictive methods to identify two cytoskeleton inhibitors that could potentially target RAB13. This research unveiled RAB13’s substantial role in modulating pathways related to cytoskeletal remodeling and tight junctions, processes well known for their influence on cell movement and migration [66,67].

Nonetheless, a significant challenge in the application of specific aptamers into future clinical therapeutics for cancer treatment results from the limited number of clinical trials dedicated to exploring these agents. Furthermore, the relatively small number of potential molecular targets for aptamers represents another problem. To overcome these challenges, there is a need to increase clinical trials for aptamer therapies in various cancers. Additionally, efforts should be directed towards identifying and characterizing a wider range of molecular targets for aptamers.

### 3.2. Aptamers for the Treatment of OC

Here, we summarized the latest aptamer reports in the literature (Figure 2), in which aptamers specifically interact with biomarkers associated with poor patient survival in OC.

#### 3.2.1. AXL Receptor Tyrosine Kinase (AXL)

Receptor tyrosine kinases (RTKs) are prominent therapeutic targets due to their pivotal involvement in regulating growth factor signaling and the metastatic behavior of tumors. AXL, a member of the receptor tyrosine kinase family, belongs to the TAM (Tyro3-AXL-Mer) receptor kinase subfamily along with its homologs, Tyro3 and Mer [68,69]. AXL has been identified as an oncogenic factor due to its ability to enhance cancer cell survival, proliferation, invasion, and metastasis. The primary ligand responsible for activating AXL is growth-arrest-specific protein 6 (Gas6), a γ-carboxylate protein that exhibits a strong affinity for the AXL receptor. Upon binding, Gas6 induces dimerization and autophosphorylation of specific tyrosine residues in the AXL receptor. Consequently, this initiates the recruitment, phosphorylation, and subsequent activation of multiple signaling pathways, including PI3K, MAPK, and PKC [70]. AXL exhibits widespread expression and has been detected in various organs and cell types, including monocytes, macrophages, and endothelial cells present in the heart, skeletal muscles, liver, and kidneys. The dysregulation of the Gas6/AXL interaction, either through overexpression or heightened activity, has been documented in different cancer types, including prostate, esophageal, thyroid, breast, lung, liver, OC, and glioblastoma, and is associated with poor prognosis. Moreover, this dysregulation has been correlated with unfavorable prognostic outcomes [68,69,71,72,73]. In a study conducted by Rankin et al., it was demonstrated that AXL exhibits elevated expression levels specifically in high-grade serous ovarian tumors and metastatic ovarian tumors [74]. However, normal ovarian epithelium and tumor stroma do not display significant AXL expression. Considering the association of AXL with prognostic indications in OC, targeting AXL in ovarian tumor cells presents a promising therapeutic approach with the potential to inhibit cancer progression. The down-modulation of AXL expression would result in a decrease in OC tumor growth and an improvement in patients’ overall survival [68,72].

Various strategies can be employed to modify aptamers, aiming to extend their half-life in serum. These modifications can be incorporated directly during the SELEX process or through post-SELEX optimizations. Given that wild-type aptamer molecules possess a notably short half-life, primarily due to their rapid clearance by the kidneys and susceptibility to nuclease-mediated degradation, their utility in physiological conditions is constrained. Consequently, a range of biochemistry-based approaches have been employed to engineer modified aptamers aimed at addressing this instability issue while optimizing the pharmacokinetic and pharmacodynamic attributes of aptamers. These specific modifications enable the targeted delivery of aptamers to precise cellular locations. These modifications involve functional optimization, multimerization, or truncation, all of which have demonstrated enhanced stability and binding efficacy. The field of aptamer technology encompasses a diverse array of modification designs, conjugation strategies, and linkage methodologies. Chemical modifications of aptamers are a common strategy utilized to bolster their stability and functionality. Notably, hydrophobic fatty acids and amphiphilic PEG components have been commonly used as agents to enhance the longevity of aptamers in drug candidates. However, it is essential to consider that high-molecular-mass PEG components have a substantial molecular weight proportion, posing a significant challenge when attempting to increase subcutaneous dosages to maximize therapeutic potential. Additionally, it is worth noting that clinical studies have reported serious or even fatal immune responses related to PEG moieties. Therefore, there are safety concerns associated with the use of high-molecular-mass PEG moieties in aptamer modifications Notably, hydrophobic fatty acids and amphiphilic PEG components have been commonly used as agents to enhance the longevity of aptamers in drug candidates. However, it is essential to consider that high-molecular-mass PEG components have a substantial molecular weight proportion, posing a significant challenge when attempting to increase subcutaneous dosages to maximize therapeutic potential. Additionally, it is worth noting that clinical studies have reported serious or even fatal immune responses related to PEG moieties. Therefore, there are safety concerns associated with the use of high-molecular-mass PEG moieties in aptamer modifications [75,76].

Kanlikilicer et al. developed a nuclease-resistant AXL aptamer to specifically target phosphor-AXL-RTK and assessed its efficacy as an antitumor agent using various intraperitoneal injection animal models. In combination with paclitaxel, the AXL aptamer demonstrated a notable augmentation of the antitumor effectiveness of paclitaxel chemotherapy. Consequently, the synergistic application of AXL-APTAMER in conjunction with chemotherapy exhibits promising potential as a strategy for addressing drug-resistant OC. Moreover, it offers the advantage of targeted delivery of chemotherapy specifically to tumor cells, thereby minimizing the adverse effects of chemotherapy on healthy cells and reducing chemotherapy-induced damage. Kanlikilicer et al. demonstrated the effectiveness of chemical modifications applied to aptamers, addressing inherent limitations and offering an attractive and efficient solution. Data gathered by Kalikilicer et al. highlight the successful application of 2′-fluoro and monothiophosphate modifications, which rendered the AXL-DNA-APTAMER stable for up to 24 h in high concentrations of human serum. This stability allowed for the effective inhibition of AXL phosphorylation, consequently leading to robust and prolonged suppression of downstream oncogenic signaling pathways, including those related to cancer metastasis and survival [71].

Amero et al. devised an innovative approach to enhance the stability and bioavailability of aptamers by converting RNA aptamers into modified DNA aptamers that specifically target phospho-AXL. Through a comprehensive analysis of a library containing 17 converted modified DNA aptamers, they identified two optimal aptamer candidates, namely GLB-G25 and GLB-A04, which exhibited superior bioavailability, and stability, and demonstrated potent antitumor effects in in vitro experiments. Notably, the incorporation of backbone modifications such as thiophosphate or dithiophosphate, along with covalent modification of the 5′-end of the aptamer using polyethylene glycol, resulted in optimized pharmacokinetic properties. These modifications improved the in vivo stability of the aptamers by reducing nuclease hydrolysis and renal clearance, ultimately leading to sustained and potent inhibition of AXL even at very low doses. Administration of these modified aptamers in orthotopic mouse models of OC yielded substantial reductions in tumor growth and the incidence of metastasis. This remarkable inhibition of the phospho-AXL target exemplifies the potential to enhance the specificity and bioavailability of aptamers through chemical modifications. Consequently, these findings establish a solid basis for further translational studies aiming to bring these aptamer candidates and companion biomarkers into clinical practice [68].

#### 3.2.2. Cancer Antigen 125 (CA125)

Cancer antigen 125 (CA125), also known as MUC16, is a transmembrane mucin with a substantial number of attached sugar molecules. It has a large molecular weight and is found to be excessively produced in 80% of cases of OC [77]. The concentration of CA125 in the bloodstream is routinely measured to aid in the diagnosis of OC, assess the effectiveness of treatment, and monitor disease progression [78]. Significantly elevated levels of CA125 are detected in 50% of patients at the early stage of the disease and in 85% of patients with advanced OC, establishing it as the widely accepted biomarker for OC in clinical practice [79,80,81]. The CA125 antigen is typically distributed throughout various anatomical sites in the human body. It is detected in the cervical mucus of healthy women and is likely synthesized and released by endocervical cells. The amniotic fluid and chorionic membrane of the developing fetus contain abundant quantities of CA125. Additionally, CA125 expression is observed in human milk, epithelial cells lining the airways, respiratory glands, and bronchial mucus. Furthermore, CA125 is expressed in diverse tissues, including the endocervix, endometrium, pleura, pericardium, peritoneum, secretory mammary glands, apocrine sweat glands, intestines, lungs, and kidneys. MUC16 (CA125) actively participates in the development of ovarian tumors and the formation of metastases by binding to the peritoneal mesothelium. In addition to its involvement in metastasis, MUC16 hinders the destruction of tumor cells by NK cells. Moreover, MUC16 obstructs the recognition of cancer cells by NK cells, consequently promoting the survival of these cells [77,79,80,81,82].

Scoville et al. employed the One-Pot SELEX method to select two aptamers, CA125_1 and CA125_12, both generated without the use of primers. These aptamers demonstrated the ability to bind to clinically relevant concentrations of the target protein. Interestingly, the presence of Mg^2+^ ions exerted distinct effects on the binding behavior of the aptamers. Specifically, the binding of CA125_1 required the presence of Mg^2+^ ions, whereas the presence of these ions abolished the binding of CA125_12. In summary, the One-Pot SELEX approach proved to be a promising selection method, resulting in the identification of DNA aptamers targeting a protein of significant clinical importance [81].

Lamberti et al. successfully isolated two RNA aptamers, CA125.1 and CA125.11, specifically targeting the CA125 tumor marker. The anti-CA125 aptamers were obtained using a protein/SELEX strategy, involving the incubation of a library with purified His-tagged CA125 protein. The transcription process was carried out in the presence of 20-F-Py, ATP, GTP, dithiothreitol, RNase inhibitors, inorganic pyrophosphatase, and a modified variant of T7 RNA polymerase, employed specifically to enhance yield. In the process, 20F-Py RNAs were employed to increase the aptamers’ resistance against degradation by nucleases. RT-qPCR analysis demonstrated that CA125.1 exhibited significantly superior binding properties compared to CA125.11. This finding was further confirmed through surface plasmon resonance (SPR) binding experiments, where the ligand-protein was immobilized on a chip, and both aptamers were tested for their response in terms of response units (RU). Further investigation focused on CA125.1 as the analyte in solution, leading to the determination of initial kinetic parameters. These parameters indicated a dissociation constant in the nanomolar range. This outcome suggests that the selected aptamer, CA125.1, holds promise as a potential diagnostic tool, particularly as a bioreceptor in SPR-based biosensors, warranting further development [82].

Gedi et al. presented a method for selecting an anti-CA125 aptamer and its subsequent utilization in a biochip featuring a three-dimensional network of carbon nanotubes (3DN-CNTs) for CA125 detection. The aptamer, named rCAA-8, exhibited a strong binding affinity for the repetitive domain of CA125, enabling differentiation between CA125-expressing cells (OVCAR-3) and CA125-negative cells (SKOV-3) through fluorescence imaging. In comparison to conventional enzyme-linked immunosorbent assays (ELISAs) targeting CA125, the aptamer-based CA125 assay performed on the 3DN-CNTs demonstrated enhanced sensitivity and a wider dynamic range based on the concentration of CA125. This improvement was attributed to the aptamer’s high specificity for the target and the high density of target molecules on the biochip surface. Consequently, the selected aptamer and its biochip applications have promising potential for facilitating the sensitive detection of CA125 in clinical settings [54].

Tripathi et al. conducted a study in which they developed a new DNA aptamer using a Membrane-SELEX approach. Among the aptamers generated, Apt 2.26 showed the highest affinity and specificity for CA125. To characterize its binding properties, membrane-based assessment was employed to observe the binding of Apt 2.26 to CA125. In vitro methods including DOT ELASA, NALFA, and DPV were utilized to investigate the binding interaction between Apt 2.26 and CA125, demonstrating the diagnostic potential of this screened aptamer. Furthermore, the stability of Apt 2.26 in human serum and high salt concentrations confirmed its robust nature, making it suitable for use with complex sample matrices [83].

#### 3.2.3. Human Epididymis Protein 4 (HE4)

Human epididymis protein 4 (HE4), initially identified in the human epididymis, is a 25 kDa serine protease inhibitor that is expressed in the reproductive and respiratory tracts. This protein exhibits interactions with various other proteins, including MUC16 (CA125) and WFDC members like SPINT4 (serine peptidase inhibitor, Kunitz type 4). HE4 is secreted into the bloodstream as a glycoprotein and is found at elevated levels in patients with serous and endometrioid EOC. Notably, HE4 is up regulated in OC, and its elevated levels have been observed in serous, endometrioid, and clear-cell ovarian tumors. Furthermore, HE4 has shown increased expression in patients with mucinous ovarian tumors, which distinguishes it from CA125 [84,85,86,87,88]. In contrast to CA125, HE4 offers several advantages as a biomarker. HE4 serum levels do not increase during pregnancy, menstruation, or in benign gynecological conditions. Additionally, HE4 is elevated at an earlier stage in the progression of the disease [84,89]. Nevertheless, the concentration of HE4 is substantially influenced by age [86].

Eaton et al. conducted a study to investigate aptamers that specifically target HE4. The research employed capillary electrophoresis and encompassed five rounds of selection using a single-stranded DNA (ssDNA) library with a 25-nucleotide random region. The ssDNA library was exposed to recombinant glutathione-S-transferase (GST)-HE4 protein. To eliminate aptamers binding to the protein tag, two rounds of counter-selection against GST were performed. Following PCR amplification, the ssDNA oligonucleotides were regenerated using streptavidin columns. Each round of selection was sequenced, and the data were analyzed using a bioinformatics data pipeline called the Rebecca Whelan Python Enrichment. Candidate aptamers were selected for further characterization based on high-fold enrichment and cluster abundance. Notably, aptamers A1, A3, and B10 exhibited affinity towards HE4 [55]. Unexpectedly, the A1 aptamer designed to bind with HE4 has been employed as a recognition probe in aptasensors specifically targeting CA125 [90,91]. In a recent advancement, the A1 aptamer sequence was utilized to create an aptasensor that utilizes up-conversion luminescence resonance energy transfer (LRET) for the detection of HE4 [92].

Hanžek et al. successfully identified novel DNA aptamers, namely AHE1 and AHE3, which demonstrate a high binding affinity to the human HE4 protein present in urine samples. These anti-HE4 aptamers were specifically selected through the implementation of Hi-Fi SELEX, a selection technique that relies on digital droplet PCR partitioning and sensitive amplification methods to isolate rare aptamer sequences. By utilizing this approach, the researchers were able to isolate aptamers that effectively target the HE4 protein within the complex urine environment. Given that HE4 levels exhibit elevation in the urine of patients, the aptamers underwent further characterization specifically in urine samples. The results revealed that AHE1 and AHE3 exhibited a favorable affinity towards the target HE4, as evidenced by a dissociation constant measured within the nanomolar range. These compelling findings strongly indicate the potential of AHE1 and AHE3 as promising diagnostic probes, which could be further harnessed in the development of advanced diagnostic tests and biosensors intended for the reliable detection of OC [84].

#### 3.2.4. Epithelial Cell Adhesion Molecules (EpCAM)

Epithelial cell adhesion molecules (EpCAM or CD326) are glycosylated membrane proteins that belong to the small GA733 protein family and are characterized as type I transmembrane glycoproteins. They have been found to be overexpressed in more than 70% of OC cases, and their expression levels are closely linked to the presence of malignant ascites, chemoresistance, and reduced survival rates among OC patients [93,94,95]. While EpCAM is typically expressed in normal epithelial tissues, its expression within the peritoneal cavity seems to be specific to tumors, as mesothelial cells in the abdominal cavity do not exhibit EpCAM expression. EpCAM plays a pivotal role in driving cancer progression by regulating various cellular attributes, including proliferation, stemness, mobility, and resistance to chemotherapy and radiotherapy. Additionally, the aberrant expression of EpCAM can influence the immune microenvironment, leading to modulations in immune responses [94,96,97]. It is important to note that the functions of EpCAM are highly reliant on the specific context and exhibit dynamic characteristics. The downregulation of EpCAM has been associated with the inhibition of cell-cell adhesion and the process of epithelial-mesenchymal transition (EMT), suggesting its involvement in the progression of OC [93,98]. Recent studies have linked EpCAM to oncogenic features, including the enhanced transcription of c-Myc and cyclin A/E. Furthermore, cells expressing EpCAM are known to possess tumor-initiating potential, making EpCAM a significant marker for OC stem cells. These findings provide compelling evidence that EpCAM represents a highly promising and suitable therapeutic target for OC [93,94,95,96,97,98,99].

Zheng et al. developed a bispecific aptamer designed to simultaneously target EpCAM and CD44, to inhibit cell growth and induce apoptosis in intraperitoneal OC cells. To achieve this, they utilized a double-stranded RNA adaptor to link two aptamers that specifically bind to EpCAM and CD44. This novel approach not only extended the half-life of the aptamers in circulation but also reduced renal filtration compared to using a single aptamer. Through cell-based studies and animal trials, the researchers observed promising results, indicating that the fused aptamer effectively suppressed OC cell growth and inhibited intraperitoneal tumor progression to a greater extent than using CD44 and EpCAM aptamers individually or in combination [93].

#### 3.2.5. CD44

CD44, also known as HCAM, Hermes antigen, or lymphocyte homing receptor, is a cell surface glycoprotein that plays a crucial role in mediating cellular responses to the surrounding microenvironment. It is involved in diverse intracellular processes including cell proliferation, survival, motility, and differentiation. Aberrant expression of CD44 has been observed in various epithelial malignancies, including head and neck, colon, and endometrial cancers as well as OC [100,101,102]. Extensive research has focused on investigating the correlation between CD44 expression or its isoforms and the survival of OC patients [102]. CD44 is found to be expressed in a majority of EOC tumors, and higher levels of CD44 are associated with advanced disease stages. Activation of the PI3K/Akt and MAPK pathways, facilitated by CD44, may inhibit apoptosis and promote invasion and metastasis of cancer cells. Moreover, CD44 levels serve as effective markers for diagnosing OC and predicting clinical outcomes. Importantly, the upregulation of CD44 in OC is strongly linked to the occurrence of metastasis and disease relapse [100,101,102,103,104].

The CD44-EpCAM aptamer, developed by Zheng et al., represents a noteworthy CD44-targeting aptamer with the ability to simultaneously bind to EpCAM. This bispecific aptamer demonstrated notable superiority in suppressing the growth of intraperitoneal tumors compared to single aptamers. The augmented efficacy of the bispecific CD44-EpCAM aptamer is primarily attributed to its prolonged circulation time in comparison to the individual aptamers. Importantly, the researchers also determined that the bispecific CD44-EpCAM aptamer exhibited an excellent safety profile as it displayed no signs of toxicity to the host. Furthermore, it was found to be incapable of eliciting innate immunogenic responses, indicating its potential suitability for therapeutic applications without triggering unwanted immune reactions [93].

#### 3.2.6. CD5 Antigen-Like Precursor (CD5L)

CD5 antigen-like precursor (CD5L), also known as apoptosis inhibitor of macrophages (AIM), is a 40 kDa soluble glycoprotein that belongs to the scavenger receptor cysteine-rich (SRCR) superfamily. This versatile protein regulates macrophage activity in a broad range of contexts, contributing to the pathogenesis of various infectious and inflammatory conditions. Furthermore, CD5L is implicated in diverse cellular processes, including atherosclerosis and cancer. Additional roles of CD5L have been discovered since, but those related specifically to endothelial cells and angiogenesis remain unknown [105,106].

LaFargue et al. presented findings that provide evidence for the involvement of CD5L in the development of adaptive resistance to bevacizumab, an antiangiogenic therapy. They demonstrated that neutralizing CD5L using either an antibody or an aptamer effectively inhibited the adaptive resistance to antiangiogenic treatment. The S76.T RNA-aptamer, developed by the researchers, was found to reduce the expression of pAKT and significantly inhibit tube formation and cell migration in AVA-resistant RF24 endothelial cells. To assess the impact of S76.T on adaptive resistance in vivo, SKOV3ip1 OC cells were injected into the peritoneal cavity of nude mice. Mice treated with S76.T aptamer alone exhibited a reduction in tumor burden, while mice treated with a combination of the aptamer and B20—an anti-VEGF (vascular endothelial growth factor) antibody—showed even greater reductions in tumor weight, fewer tumor nodules, as well as lower microvessel density and proliferation compared to mice treated with a control scramble IgG. These collective findings highlight CD5L as a significant protein involved in the adaptive response to anti-VEGF treatment. Consequently, targeting CD5L could be a beneficial strategy for patients undergoing treatment with antiangiogenic drugs [106].

#### 3.2.7. Vimentin

Vimentin, an extracellular matrix protein belonging to the intermediate filament protein family, has garnered significant attention in cancer research. Numerous studies have demonstrated a direct association between elevated vimentin expression and reduced survival rates across various cancer types, including colorectal, cervical, breast, gastric, and non-small-cell lung cancers [107,108,109]. The presence of vimentin filaments within cancer cells serves as a protective mechanism during migration and traversal through narrow spaces, as they establish a viscoelastic framework that can withstand mechanical stresses. Furthermore, vimentin plays a crucial role in maintaining the structural integrity of organelles, particularly the nucleus, during the process of EMT and cancer progression [110]. Notably, vimentin has been found to safeguard cancer cells against internal stress induced by misfolded proteins by directly binding to stress granules and aggresomes, facilitating their subsequent degradation [111]. Moreover, vimentin overexpression has been linked to an augmented metastatic capacity attributed to the induction of EMT in ovarian tumor cells [107]. Apart from its structural functions, vimentin also participates in various signaling pathways associated with cell adhesion, migration, and apoptosis. However, limited research has explored the specific relationship between vimentin expression and ovarian tumor prognosis [108,110,111,112]. Szubert et al. demonstrated that increased vimentin expression in ovarian tumor cells was correlated with a prolonged overall survival rate [113].

Costello et al. have discovered two distinct truncated aptamer motifs, namely V3M2 and V5M2, which exhibit effective binding to the extracellular matrix protein vimentin in OC cells and human ovarian tumor tissue. By truncating specific regions within the stem-loop structure, the binding affinity of the aptamers to the vimentin protein was significantly enhanced compared to that of the full-length aptamer sequences V3 and V5 [107].

#### 3.2.8. Nucleolin

Nucleolin also referred to as C23, is a prominent protein abundantly present in the nucleolus, constituting approximately 10% of the protein content within this subnuclear compartment [114]. Evidence has established the involvement of nucleolin in crucial cellular processes, including nucleolar chromatin remodeling, pre-RNA maturation, rDNA transcription, and ribosome assembly [115,116]. In normal cells, nucleolin, characterized by its RNA recognition motifs, predominantly resides within the nucleolus. However, in cancer cells, a remarkable upregulation of nucleolin has been observed on the cell surface [117]. This abnormal overexpression of nucleolin is closely associated with adverse patient outcomes, as this protein exerts oncogenic effects by promoting carcinogenesis, cellular proliferation, metastasis, and angiogenesis. Notably, nucleolin undergoes extensive phosphorylation, which is tightly regulated throughout the cell cycle. Although no mutations or splicing variants of nucleolin have been linked to malignancy, dysregulated accumulation of nucleolin messenger RNA (mRNA) and/or protein has been reported in various cancer types [114,115,116].

Ruan et al. successfully assembled six AS1411 aptamers onto two hexangular DNA tiles, resulting in the creation of two distinct aptamers with efficient and divergent structures: HA-6AS, characterized by a three-dimensional tubular shape, and ST-6AS, possessing a two-dimensional six-star configuration. Their investigation revealed that both HA-6AS and ST-6AS exhibited superior capabilities in loading doxorubicin (DOX) compared to DNA tetrahedron structures. Moreover, these aptamers effectively delivered DOX into OC cells, enabling its evasion from lysosomes and subsequent localization within the cell nucleus. HA-6AS-DOX and ST-6AS-DOX demonstrated rapid cytotoxicity specifically against OC cells while exhibiting low levels of cytotoxicity towards normal ovarian epithelial cells. Additionally, the study revealed contrasting internalization patterns between HA-6AS and ST-6AS; HA-6AS displayed a more active targeting mechanism, whereas ST-6AS exhibited a predominantly passive targeting mechanism [118].

#### 3.2.9. Stress-Induced Phosphoprotein 1 (STIP1)

STIP1, also referred to as Hsp70/Hsp90-organizing protein (HOP), encompasses three tetratricopeptide repeat domains that enable interactions with heat shock proteins (Hsps), leading to the formation of complexes involved in various biological processes [119]. These processes encompass RNA splicing, transcription, protein folding, signal transduction, and cell cycle regulation. STIP1 serves as a mediator for protein transfer between Hsp70 and Hsp90, thus facilitating the formation of the Hsp70/Hsp90 complex. This complex plays a critical role in the folding and maturation of transcription factors, hormone receptors, and protein kinases. Hsps are implicated in the initiation, facilitation, and progression of malignant conditions [120,121,122]. Reports have indicated the overexpression of STIP1 in ovarian, breast, renal, gastric, and colorectal cancers and oral squamous cell carcinoma [121,122]. In human OC, STIP1 has been identified as a released biomarker. In the context of human OC cells, secreted STIP1 has been found to enhance tumor cell proliferation by binding to ALK2 (activin A receptor, type II-like kinase 2) and activating the SMAD-ID3 signaling pathways. Notably, clinical investigations have revealed that elevated expression of the STIP1 protein is associated with unfavorable prognostic outcomes in cases of OC [120].

Van Simaeys et al. presented a novel approach for the straightforward identification of the target protein of aptamer 1, which was found to be STIP1. They further revealed the involvement of STIP1 in the invasiveness of the TOV-21G OC cell line. Building upon the identification of STIP1 as a promising biomarker for OC through their methodology, the researchers subsequently explored the therapeutic potential of aptamer TOV6. Remarkably, the aptamer exhibited the capability to inhibit cellular invasion, thus highlighting its potential as a valuable tool for therapeutic intervention [59].

Chen et al. presented a novel approach for the early diagnosis of OC, involving the respective and simultaneous detection of CA125 and STIP1. They utilized aptamer-based fluorescent and RLS (resonance light scattering) sensors, which provided strong evidence for the presence of OC in its early stages. The findings demonstrated that these sensors exhibited excellent sensitivity and specificity in detecting the tumor markers CA125 and STIP1, thereby offering a promising diagnostic tool for OC [123].

#### 3.2.10. Heat Shock Proteins Hsp70 (HSP70s)

The 70 kDa heat shock proteins (HSP70s) constitute a highly conserved and inducible group of heat shock proteins. Their primary role lies in acting as molecular chaperones and participating in various cellular processes that involve protein folding and remodeling [124,125,126]. HSP70s are frequently observed to be overexpressed and can serve as prognostic markers in numerous cancer types [126]. Moreover, they play critical roles in the molecular processes underpinning cancer hallmarks, as well as influencing the growth and survival of cancer cells. Interestingly, the impact of HSP70s on cancer cells extends beyond their chaperone activities; they also exert significant regulatory effects on cancer cell signaling. Consequently, HSP70′s high expression in cancer cells is often associated with unfavorable outcomes, including poor prognosis, disease progression, recurrence, and resistance to treatment [124,125,126,127,128].

Rérole et al. have discovered a range of peptide aptamers capable of binding to HSP70. In the context of human tumor cells, two of these peptide aptamers, namely A8 and A17, exhibited binding affinity to distinct regions of HSP70—the peptide-binding and ATP-binding domains, respectively. Remarkably, these aptamers effectively impeded the chaperone activity of HSP70, thereby heightening the susceptibility of cells to apoptosis induced by anticancer drugs. Significantly, P17, a 13-amino acid peptide derived from the variable region of A17, exhibited persistent specific inhibitory characteristics targeting HSP70. Interestingly, in vivo investigations employing both local and systemic administration of P17 demonstrated substantial regression of subcutaneous tumors. Notably, this therapeutic outcome displayed a noteworthy association with a pronounced influx of macrophages and T lymphocytes into the TME [125].

#### 3.2.11. Mucin 1 (MUC1)

MUC1, also known as episialin or EMA, is a prominent member of the mucin family and functions as a transmembrane glycoprotein. Structurally, it forms a type-I transmembrane heterodimer consisting of two subunits that are non-covalently bound together [129,130]. In normal physiological conditions, MUC1 is primarily expressed on the apical surface of secretory epithelia, including those found in the mammary gland, gastrointestinal tract, respiratory system, urinary system, and reproductive organs [131]. However, upregulated expression of MUC1 contributes to the advancement of tumors through its influence on multiple signaling pathways as well as its regulation of tumor cell proliferation and epithelial-mesenchymal transition [132,133,134]. In the context of cancer progression and metastasis, the expression of MUC1 is characterized by elevated levels, altered glycosylation patterns, and abnormal distribution on the cell surface [135,136]. Notably, the MUC1 oncogene is frequently overexpressed in various epithelial adenocarcinomas such as lung, liver, pancreatic, and breast cancer and OC [132,137].

Ferreira et al. have successfully engineered a collection of DNA aptamers capable of binding to MUC1 peptides with exceptional affinity and selectivity. Among the assortment of aptamers generated during their investigation, S2.2 exhibited the most favorable K_d_, implying a superior binding affinity towards the MUC1 target. Additionally, the researchers conducted a thorough evaluation of the aptamers’ diagnostic assay potential, thereby highlighting their prospective utility in clinical diagnostic applications [53]. In their subsequent investigation, the research team devised aptamer 5TR1, which selectively targets the five consecutive repeats located within the variable tandem repeat region of MUC1. Subsequently, this aptamer was employed in the creation of a hybrid sandwich ELISA in conjunction with an antibody. This innovative approach facilitated the identification and quantification of MUC1 in buffered solutions. Furthermore, recovery studies conducted under buffer conditions yielded averaged recoveries ranging from 98.2% to 101.7% for all spiked samples, thus substantiating the aptamer’s viability as a receptor in microtiter-based assays. Notably, the developed aptamer exhibited comparable binding characteristics to the corresponding antibody and even competed with the antibody for binding [138].

#### 3.2.12. Human Epidermal Growth Factor Receptor 2 (HER2)

Human epidermal growth factor receptor 2 (HER2), also known as Neu or ErbB2, belongs to the family of epidermal growth factor receptors (EGFRs) that are receptor tyrosine kinases. EGFRs play crucial roles in regulating cell proliferation and differentiation during both embryonic development and adult tissue homeostasis [139]. In numerous cancer types, HER2 exhibits overexpression and is associated with unfavorable prognosis, suboptimal treatment outcomes, and reduced survival rates [140,141,142]. Correspondingly, overexpression of HER2 in human OC cell lines has been linked to enhanced DNA synthesis, accelerated cell growth, improved efficiency in soft agar cloning, and increased tumorigenicity in xenograft models using nude mice [139,143]. Various HER2-targeted therapies have been developed to treat HER2-overexpressing tumors, effectively downregulating HER2 expression. Consequently, profiling HER2 expression is of significant importance in cancer prognosis, patient stratification, and monitoring therapy response [139,140,141].

Varmira et al. developed a modified RNA aptamer conjugated with hynic and labeled with 99m Tc, aiming to create a radiopharmaceutical suitable for diagnostic imaging of OC cells (SKOV-3) exhibiting high expression of HER2. The complex demonstrated stability in normal saline and serum while specifically targeting the HER2 receptor on cancer cells. Upon injection of the ^99m^Tc-hynic-RNA aptamer, rapid clearance from the bloodstream was observed, with predominant excretion occurring through the hepatobiliary system. However, despite the in vitro specificity of the radioconjugated aptamer in binding to the HER2 receptor on cells, no significant tumor-to-blood or tumor-to-muscle ratios were observed. This lack of specificity binding may be attributed to the in vivo biodistribution of the ^99m^Tc-hynic-RNA aptamer. Based on these findings, the team concluded that modifying the radiometal chelator and the structure of the RNA aptamer could potentially improve tissue uptake and enhance the pharmacokinetic properties of the radiolabeled aptamer for improved tumor targeting [140].

Zhu et al. employed a combinatorial screening approach, both in vitro and in vivo, to identify DNA aptamers targeting HER2 receptors. These aptamers, named Heraptamers, were subsequently labeled with ^18^F to enable their utilization in positron emission tomography (PET) imaging of HER2 in OC. Initially, the Heraptamers were selected and validated through in vitro assessments involving the HER2 extracellular domain (ECD) and HER2-positive SKOV-3 OC cells. Subsequently, the aptamers were modified with an alkyne, radiolabeled with ^18^F using azide-functionalized precursors through click chemistry, and evaluated in SKOV3-tumor-bearing mice using PET imaging. Notably, two aptamers, Heraptamer1 and Heraptamer2, exhibited rapid and significant accumulation in tumors within a short timeframe of just 1 h. In contrast, these aptamers demonstrated minimal uptake in control tumors, underscoring their specific targeting capability [141].

#### 3.2.13. Programmed Death-Ligand 1 (PD-L1)

Programmed death-ligand 1 (PD-L1) is a transmembrane protein that serves as a major ligand for PD-1. It is characterized by its overexpression in various cancer cells and is recognized as a potential biomarker in cancers exhibiting elevated PD-L1 levels. PD-L1 plays a critical role in modulating the host’s antitumor immune response, enabling tumor cells to evade immune surveillance and promoting metastasis [57,144,145]. Despite the significance of PD-L1 expression as a prognostic indicator, the immunological pathway associated with high PD-L1 expression in OC remains understudied, and the molecular mechanisms involving immune cells and tumor cells remain elusive [145,146].

Yazdian-Robati et al. have presented novel DNA aptamers that exhibit highly specific and strong binding affinity towards PD-L1. The researchers evaluated the affinity and specificity of two screened aptamers, namely Apt5 and Apt33. Furthermore, they examined the internalization of Apt5 for potential applications in targeted delivery and imaging. Aptamer Apt5 demonstrated exceptional affinity and selectivity for PD-L1 and was observed to be internalized into OC cells. Additionally, the results indicated that Apt5 successfully detected cancer cells, thus highlighting its potential for early-stage cancer cell detection [57].

#### 3.2.14. Kisspeptin-1 (KiSS-1)

Kisspeptin, a polypeptide, and its corresponding encoding gene KiSS-1 were initially identified as an inhibitor of human metastasis and as a gene associated with malignant melanoma [147]. Intriguingly, recent research has indicated significant alterations in the expression of kisspeptin and its receptor in women affected by various cancer types, including lung cancer, OC, breast cancer, and endometrial cancer. In recent years, extensive investigations have explored the roles of KiSS-1 and KiSS-1R in diverse malignant tumors, highlighting their potential as prognostic or therapeutic markers in cancers [147,148]. The molecular mechanisms underlying the regulation of cancer cell proliferation and metastasis by the kisspeptin/KiSS-1R system involve intricate endocrine, paracrine, and autocrine actions. Additionally, this system may play a role in modulating tumor tissue angiogenesis. Consistent with these findings, a recent study demonstrated higher preoperative expression of KiSS-1 in ovarian tumor tissue compared to non-tumor tissue. Moreover, the presence of metastasis and tumor size exhibited a negative correlation with preoperative KiSS-1 expression [147,149,150,151].

Singh et al. focused on the utilization of KiSS-1 as a target for novel and specific Kisspeptin aptamers. The researchers concluded that for diagnosis purposes, it is likely that a combination of kisspeptin levels with established tumor markers would be required, as a singular measurement of kisspeptin may be insufficient. Additionally, they suggested that the development of specific aptamers against the KiSS1R holds potential and could aid in excluding biological interactions with other receptors, such as NPFF1 and NPFF2, which also respond to kisspeptin. The use of highly selective aptamers with a strong affinity for kisspeptin would enable more precise targeting of this molecular target within cancer-related pathophysiological mechanisms. This approach would minimize the risk of nonspecific blockade of other chemical mediators or interference with different signaling pathways involving kisspeptin, thereby reducing the likelihood of adverse effects in patients. Consequently, the application of such specific aptamers in therapeutic strategies for OC holds considerable advantages in the clinical management of these conditions, emphasizing the need for further research in this area [152]. Table 1 provides an overview of aptamers, target molecules, and the specific cancer types in which they exhibit activity.

#### 3.2.15. Ephrin-A2 (EphA2)

The erythropoietin-producing hepatocellular receptors (EPHs) represent the largest subfamily of receptor tyrosine kinases, engaging with membrane-bound proteins known as ephrins [153]. EPHs and their ligands are widely expressed, particularly during early development, across various cell types, participating in several physiological functions critical for embryonic development, including the regulation of processes such as cell migration and adhesion [154,155,156]. In contrast to other EPH kinases, EphA2, a transmembrane protein with a molecular weight of 130 kDa, predominantly resides in adult human epithelial cells. Its expression is typically low [154,157]. The EphA2 receptor mediates signaling pathways that negatively modulate epithelial cell growth, suppress cell migration, and induce EphA2 internalization and degradation [158,159]. However, malignant cells often exhibit weakened cell-cell contacts, preventing the association of the EphA2 receptor with ephrin, and leading to upregulated EphA2 receptor expression. High levels of EphA2 expression have been reported in various solid tumors, including ovarian, prostate, pancreatic, lung, esophageal, colorectal, and breast cancers as well as bone sarcomas, melanoma, and glioblastoma [157]. Furthermore, numerous studies have demonstrated that the overexpression of the EphA2 receptor plays a pivotal role in promoting the aggressiveness and metastatic potential of many of these cancers [160]. Moreover, EphA2 is closely associated with key regulators of angiogenesis [161]. Hence, EphA2 represents a critically important therapeutic target, and agents that promote EphA2 activation hold significant potential for suppressing cancer cell malignancy.

Santana-Viera et al. identified a 20-fluoro-modified pyrimidine RNA aptamer named ATOP, which specifically targets EphA2. The ATOP EphA2 aptamer was discovered through the use of an aptamer clustering algorithm, which aimed to identify a common sequence-structure motif derived from two parallel selection processes: protein SELEX 48 involving hEphA2 and cell-internalization SELEX 49 involving MDA231 cells expressing EphA2. When applied to tumor cell lines expressing EphA2, the ATOP EphA2 aptamer effectively reduced tumor cell migration and clonogenicity. In an in vivo mouse model of Ewing sarcoma metastasis, the ATOP EphA2 aptamer exhibited the ability to decrease primary tumor growth and significantly reduce the occurrence of lung metastases. Consequently, the EphA2 ATOP aptamer emerges as a promising candidate for the development of next-generation targeted therapies, offering safer and more effective treatment options for EphA2-overexpressing tumors [157].

### 3.3. Aptamers Enhancing Immunity in OC

Immunotherapeutics represent a revolution in medicine. However, their clinical application is limited in OCs due to the immunosuppressive nature of the TME. Therefore, the identification of potential targets to overcome immunological tolerance and enhance the immune response is one of the challenges facing medicine. Currently, immunotherapy relies on drugs that target immune checkpoints. These approaches can be used in monotherapy as well as in combination therapy, which may include antiangiogenic agents, PARP inhibitors, cisplatin, and more. One of the innovative approaches to enhancing the immune response involves the use of aptamers [162]. Compared to antibodies, aptamers are more stable, can be modified, and can be used for targeted drug delivery [15].

Understanding the TME of OC cells allows for the development of therapies aimed at stimulating the immune system to fight cancer. Immunotherapeutic strategies are currently one of the primary research goals of scientists. The latest scientific reports indicate that myeloid-derived suppressor cells (MDSCs) participate in immunosuppression, which may be correlated with high expression of HSP70. HSP70 is a heat shock protein that participates not only in the immune response but also in regulating apoptosis, necrosis, and angiogenesis. A comparison of urine samples from healthy donors and cancer patients revealed a higher concentration of HSP70 in cancer patients, including OC patients. Consequently, Gobbo et al. developed an aptamer, A8, which inhibits the interaction between HSP70 and TLR2, thereby blocking the activation of MDSCs. It has been demonstrated that aptamer A8 leads to the inhibition of STAT3 phosphorylation and the secretion of IL-6 in a medium containing tumor-derived exosomes. In vivo studies showed that a combination of 5-Fluorouracil and cisplatin may decrease the activation of MDSCs [163]. These experiments have shown novel perspectives on the use of aptamers in the immunotherapy of OC.

T cells assume a crucial role in immune responses. Nevertheless, within the TME, they undergo differentiation, adopting an exhausted phenotype characterized by heightened expression of immune inhibitory checkpoints, including TIM-3. Significantly, TIM-3 assumes an immunosuppressive role in the context of OC. It hinders the intracellular transport of nucleic acids into ovarian tumor cells’ cysts, consequently diminishing the therapeutic efficacy of chemotherapy [164]. This elevation in TIM-3 expression holds notable significance in the pathogenesis of OC [165]. In 2017, Gefen et al. designed a TIM-3 aptamer to enhance T-cell cytotoxic activity in a mouse model injected with colon carcinoma. In vitro studies demonstrated the highest increase in the proliferation of CD8+ T cells, which was associated with an upregulation of cytotoxic cytokines, including IFN-γ and TNF-α. Moreover, in vivo studies revealed that mice treated with the aptamer targeting TIM-3 experienced a more significant reduction in tumor volume correlated with an increased number of activated T cells compared to mice treated with a TIM-3 antibody. These results indicate that aptamers may offer a promising approach to activating T cells by inhibiting the TIM-3 receptor [166].

One of the inhibitors of the immune system response is PD-L1, overexpression of which was found in cancer cells. PD-L1 exerts its inhibitory effects by engaging with the PD-1 receptor, resulting in the attenuation of cytotoxic cytokine production and, consequently, a reduction in the efficacy of T-cell activity. As outlined earlier, Yazdian-Robati et al. designed two aptamers, Apt5 and Apt33, that bind to PD-L1, offering the potential for targeted delivery of anticancer drugs and increasing the immune response. Apt5 exhibited the highest affinity for binding to PD-L1 with high specificity. Furthermore, the researchers demonstrated that labeling this aptamer with ATTO647N could be a promising approach for early-stage OC detection. It is important to highlight that this labeling procedure comes with a significant cost. Therefore, further studies are needed to optimize the production cost of the presented detection method [57].

## 4. Aptamers Conjugated to Non-Coding RNA and Nanoparticles as Therapeutics in Ovarian Cancer

Aptamers can effectively transport proteins, drugs, or nucleic acids into precise cellular structures through conjugation with non-coding RNAs, drug molecules, or NPs. This targeted delivery mechanism aids in mitigating toxicities and minimizing side effects associated with the transported substances [76]. Herein, we provide exemplars of aptamers that have been conjugated with non-coding RNA molecules and NPs.

ncRNA-based therapeutics have emerged as a significant focus in the field of cancer therapeutics [19,167]. ncRNAs have demonstrated the ability to modulate gene expression associated with metabolic disorders, genetic diseases, and cancers, leading to decreased expression levels [18,168,169,170]. Recent research has unveiled the crucial role of ncRNAs in diverse cellular processes linked to OC progression, including cell proliferation, apoptosis, invasion, migration, chemoresistance, angiogenesis, and energy metabolism reprogramming. Examining ncRNAs as prognostic biomarkers holds promise for advancing approaches for OC patients. Gaining a comprehensive understanding of ncRNA mechanisms involved in OC metastasis and chemoresistance could pave the way for novel therapeutic interventions aimed at improving patient prognosis [18,20,171,172].

Substantial strides have been made in the development of ncRNA-based drugs, as evidenced by the approval of various nucleic acid therapeutics by regulatory agencies such as the United States Food and Drug Administration (FDA) and the European Medicines Agency (EMA) [167]. Notably, in 2018, the FDA approved patisiran, the first globally approved small interfering RNA (siRNA) drug [173]. Subsequently, regulatory approvals were granted for the use of givosiran and lumasiran [174,175]. The rational design of ncRNAs, including siRNAs, small activating RNAs (saRNAs), and miRNAs, allows for the precise targeting of a diverse range of genes and proteins. Accurate identification of target cells is essential for the efficient and precise delivery of ncRNAs. Aptamers, generated through SELEX technology, exhibit high specificity and selectivity for binding to various proteins present on the cell surface. Consequently, aptamers function as molecular recognition entities, enabling precise recognition and binding to target cells, thereby facilitating efficient delivery of cargo into the cellular environment [173] siRNA and miRNA molecules are potent agents for gene silencing and have garnered significant attention as a novel class of gene-based therapeutics in cancer treatment. However, their clinical utility has been hampered by challenges related to their limited stability, vulnerability to nuclease degradation, and non-specific delivery to cells and tissues in vivo. To address this limitation, the integration of aptamers, which offer high targeting specificity, with siRNA/miRNA technology has emerged as a promising approach to achieve precise and efficient gene targeting. Through modification with aptamers, ncRNAs can be loaded into aptamer-functionalized nanosystems, enabling precise delivery to specific cells, tissues, or organs for internalization. This synergistic combination enables selective modulation of specific genes, enhancing the therapeutic potential of siRNA/miRNA-based interventions in cancer therapy [19,20,170,176,177,178,179]. Aptamer-NP and -ncRNA chimera are thought to undergo internalization via receptor-mediated endocytosis and subsequent release into the cytoplasm, where they come into contact with the RNAi machinery. It has been postulated that ncRNA exhibits the ability to escape endosomes and subsequent transit to lysosomes. This is of particular significance as lysosomes are known to harbor nucleases and maintain low pH conditions, which are prone to causing degradation of the ncRNA component [23]. Aptamer-functionalized NPs and aptamer-based chimeras have exhibited noteworthy efficacy in selectively delivering anticancer drugs, as evidenced by numerous preclinical and animal studies. However, as of now, none of these approaches have transitioned into clinical trials or practical applications. Despite their considerable potential, the path toward clinical approval of aptamer-based therapeutics entails several critical stages. These include gaining a more comprehensive insight into their intracellular behavior, refining their optimal formulation, and assessing their toxicity profile [75,180,181,182,183].

### 4.1. Aptamer-Mediated siRNA Targeted Therapeutics

RNAi is a post-transcriptional mechanism that functions by inhibiting gene expression through targeted cleavage of specific regions within a target mRNA. RNAi has exhibited significant therapeutic potential in various diseases, particularly in cancer, where it targets genes involved in well-established oncogenic pathways [17,179,184]. Numerous studies have demonstrated the efficacy of RNAi in reducing cancer cell proliferation, promoting apoptosis, and enhancing the sensitivity of resistant cancer cells to chemotherapy and radiation. Despite the development of effective siRNAs against cancer cells, there are currently no approved siRNA-based therapies for cancer treatment. The main obstacle to the successful translation of siRNAs into effective clinical therapies lies in their delivery. siRNA-based drugs, stemming from the RNAi mechanism, exhibit the ability to selectively silence genes implicated in cancer initiation, progression, and metastasis. By targeting these genes, siRNA drugs can effectively inhibit tumor cell proliferation, promote cell differentiation, suppress metastasis, and offer promising therapeutic prospects for the treatment of cancer. siRNAs offer numerous advantages in cancer therapy, including their high safety profile and efficacy [170,178,179]. Targeted delivery of siRNAs can be classified into two main categories: passive delivery, which relies on enhanced permeability and retention effects, acid pH values, and redox-triggering mechanisms to facilitate targeted uptake or transfection processes, and active delivery, where small molecules, antibodies, aptamers, or other biomolecules are employed to mediate the uptake and release of siRNAs [185]. In recent years, the utilization of nanomaterials for siRNA transfection has garnered significant interest, including lipid and polymer NPs, gold NPs (AuNPs), magnetic NPs, and mesoporous silicon NPs [186,187,188,189].

Chen et al. conducted a study aiming to suppress the expression of NOTCH3, a marker associated with OC recurrence and resistance to chemotherapy, using an aptamer-siRNA chimera delivery system based on gold NPs (Figure 3A). The aptamer-siRNA chimera was designed to specifically target OC cells through the recognition of VEGF, a protein known to be overexpressed in OC cells. The designed aptamer exhibited a positive charge of approximately +20 mV, enabling it to interact with the negatively charged cell membrane and facilitate rapid endocytosis. Notably, the aptamer demonstrated successful targeting of VEGF signaling in cisplatin-resistant OC cells. Western blot analysis confirmed the effective knockdown of NOTCH3 using the nanoparticle–chimera delivery system, surpassing the performance of a lipofectamine-mediated delivery system or siRNA alone. Furthermore, the cell viability of OC cells was significantly reduced compared to the untreated control. These findings highlight the potential of this aptamer–siRNA chimera delivery system as a promising approach for nanoparticle-mediated siRNA delivery, addressing drug resistance in OC [190].

Despite the potential enhancement of later rounds of chemotherapy through the suppression of genes associated with drug resistance in OC, an alternative and promising therapeutic approach involves the simultaneous delivery of siRNA and anticancer drugs for immediate effect. Co-delivering siRNA and anticancer drugs hold the potential to augment the effectiveness of chemotherapy by concurrently triggering apoptosis and inhibiting pro-tumor factors such as Bcl-2. Notably, cisplatin, doxorubicin, and paclitaxel have been widely studied as major chemotherapeutic agents in combination with siRNA delivery. Nonetheless, the efficacy of these anticancer drugs might be limited by proteins that disrupt their intended pathways. To counteract this resistance, siRNA has been employed to reduce cell viability by halting cell division during mitosis. Encapsulating chemotherapy agents and siRNA within NPs enables targeted delivery to cancer cells, thereby suppressing drug resistance mechanisms and promoting apoptosis [21].

He et al. conducted a research study to investigate combination therapies targeting drug resistance in OC. They employed a nanoscale coordination polymer (NCP) system for the simultaneous delivery of cisplatin and specific siRNA molecules. The siRNA molecules were designed to target mRNA encoding survivin (BIRC5), the apoptosis regulator BCL2 (BCL2), and P-glycoprotein (P-gp or ABCB1), all associated with the overexpression of multidrug resistance (MDR) genes. These targeted siRNAs were attached to the surface of self-assembled NCPs composed of metal ions and organic bridging ligands. The delivery system demonstrated remarkable efficacy in enhancing the cellular uptake of cisplatin, exhibiting a 250% increase compared to free cisplatin. Following a 24 h treatment of OVCAR-3 and SKOV-3 cells with the NCP, the cells exhibited a 50% higher rate of apoptosis compared to those treated with free cisplatin alone. Furthermore, subsequent in vivo investigations using SKOV-3 subcutaneous xenografts in mice revealed a substantial reduction in average tumor volume, approximately 2.5 times smaller, after 28 days of therapy with the NCP system. These findings indicated that the therapy was more effective than the control treatments [191].

Kotcherlakota et al. developed a gold nanoparticle-based system, denoted as TDDS (Au-TR-DX-siERBB2), for the co-delivery of doxorubicin and *ERBB2* siRNA to SKOV-3 cells. The *ERBB2* gene, which induces the production of HER2, was selected due to its association with heightened cell proliferation rates and overexpression in SKOV-3 cells and various other cancer cell types. In vivo investigations employing mice harboring SKOV-3 tumors revealed that the TDDS nanoparticle system exhibited superior efficacy in restraining tumor growth compared to control groups that included doxorubicin, siRNA, or the co-delivery of doxorubicin and siRNA without a nanoparticle carrier. Moreover, the biodistribution analysis of TDDS validated the selective accumulation of gold NPs within tumor tissues, thus solidifying the effectiveness of this nanomedicine-based combination strategy for OC. Additionally, toxicological assessments confirmed that the co-administration of doxorubicin and *ERBB2* siRNA exhibited minimal off-target toxicity due to the specific targeting of HER2 [192].

Salzano et al. synthesized a self-assembling polymeric micelle composed of PEG 2000-phosphatidyl ethanolamine (PEG2000-PE). This micelle effectively encapsulated both survivin siRNA (siBIRC5) and paclitaxel, and its therapeutic potential was assessed in vivo using mice bearing SKOV-3 tumors. The findings from this study demonstrated a substantial decrease in survivin expression within tumor tissues accompanied by potent anticancer activity when the nanoparticle formulation was administered. In contrast, the same dose of free paclitaxel did not exhibit comparable effects on the tumors. The co-delivery of survivin siRNA and paclitaxel resulted in the highest level of tumor cell apoptosis compared to the administration of paclitaxel or survivin siRNA alone. The co-delivered nanoparticle significantly reduced survivin expression. This research concluded that the inclusion of siBIRC5 along with paclitaxel sensitizes cancer cells to lower doses of chemotherapy, particularly in comparison to previously ineffective doses of paclitaxel alone [193].

### 4.2. Aptamer-Mediated miRNA Targeted Therapeutics

The advancement of research has led to the utilization of miRNA molecules for the regulation of gene expression at the mRNA level, thereby facilitating the development of miRNA-based therapeutics for a wide range of diseases [17,185]. Dysregulation of miRNA production can have profound implications for cellular fate [194]. It is well established that miRNAs can exhibit either upregulation or downregulation in various human cancers. Overexpressed miRNAs often function as oncogenes by suppressing tumor suppressor genes, whereas downregulated miRNAs can act as tumor suppressors by negatively regulating oncogenes. Numerous miRNAs have been associated with OC prognosis [20,179]. Emerging evidence highlights the crucial roles of miRNAs in pathways linked to resistance against anticancer drugs such as cisplatin and microtubule-targeting agents. However, targeted delivery of miRNAs remains a significant challenge. In recent years, diverse approaches have been employed to design aptamer-based miRNA conjugates, demonstrating promising outcomes in various diseases. Similar to siRNA, specific aptamers can selectively deliver miRNAs to desired tissues. Current research efforts are focused on developing miRNA nano-formulations that enhance cellular uptake, bioavailability, and accumulation at tumor sites, aiming to overcome delivery barriers and optimize therapeutic efficacy [16,20,22,170,194].

Dai et al. developed a chimera named Chi-29b, comprising a MUC1 aptamer that specifically targeted the MUC1 protein on the surface of tumor cells, along with miR-29b, which inhibited the expression of DNA methyltransferases (DNMTs) and subsequently restored the expression of the *PTEN* gene (Figure 3B). The researchers analyzed the specificity and effectiveness of the chimera’s delivery in OVCAR-3 ovarian tumor cells, and its biological activities were assessed by examining the expression of downstream molecules and cell apoptosis. Results indicated that the Chi-29b chimera exhibited preferential internalization into OVCAR-3 cells expressing MUC1. This led to a decrease in the expression of DNMTs, which are specific targets of miR-29b, and subsequently restored the expression of *PTEN*. Additionally, the chimera induced apoptosis in the cells [194]. In a subsequent study by the same research group, it was determined that intratumoral injection of the Chi-29b chimera significantly suppressed the growth of xenograft OVCAR-3 tumors. This effect was achieved by downregulating PTEN methylation, leading to subsequent *PTEN* expression, as well as downregulating the expressions of MAPK 4 and IGF1. The administration of Chi-29b through intraperitoneal injection resulted in a significant augmentation of apoptosis in OVCAR-3 cells that were resistant to paclitaxel. Moreover, it effectively inhibited the growth of xenograft OVCAR-3-Taxol tumors. The ability of Chi-29b to counteract chemoresistance in OVCAR-3-Taxol tumors was attributed to the activation of PTEN signaling and the subsequent downregulation of MAPK 4 and 10, as well as the expression of IGF1 [195].

Liu et al. investigated the chimera consisting of the MUC1 aptamer and let-7i miRNA in OVCAR-3 OC cells. The results obtained from their study demonstrated the specific delivery of the chimera into OVCAR-3 cells, and the subsequently released let-7i miRNA significantly enhanced the cell’s sensitivity to paclitaxel, leading to inhibition of cell proliferation, induction of apoptosis, and reduced long-term cell survival in OC cells. The chimera effectively reversed chemoresistance by downregulating the expressions of cyclin D1, cyclin D2, Dicer 1, and PGRMC1. These findings indicated that the MUC1/let-7i chimera holds the potential to specifically reverse chemoresistance to paclitaxel in OC cells [196].

Li et al. discovered that miR-203b-3p is significantly downregulated in advanced OC. Furthermore, their study demonstrated that miR-203b-3p exhibits potent inhibition of OC cell invasion and the formation of peritoneal metastases. To elucidate the molecular mechanism behind the invasion-suppressive effect of miR-203b-3p, the researchers discovered that miR-203b-3p exerts its inhibitory action on CXCL1 by unconventionally targeting CXCL1 mRNA. The binding interaction occurs between the 5′-untranslated region (5′-UTR) and protein-coding region of CXCL1 mRNA and specific regions of miR-203b-3p that are distinct from the conventional nucleotides found in the seed sequence. The researchers observed that the reintroduction of recombinant CXCL1 significantly reversed the suppressed invasive behaviors of OC cells induced by miR-203b-3p. This finding strongly suggests that miR-203b-3p hinders OC cell invasion by reducing the abundance of CXCL1. In their study, the researchers demonstrated the effective delivery of miR-203b-3p into OC cells using the EpCAM aptamer. They further showed that the EpCAM aptamer-mediated delivery of miR-203b-3p significantly inhibited the formation of peritoneal metastases and prolonged the lifespan of mice-bearing tumors. In conclusion, the study reveals a novel mechanism in which reduced expression of miR-203b-3p leads to the maintenance of elevated CXCL1 levels, thereby promoting the progression of OC. Significantly, the findings propose that the delivery of miR-203b-3p using the EpCAM aptamer holds therapeutic potential for targeting advanced OC [197].

### 4.3. Aptamer/Anti-miR Targeted Therapeutics

Anti-miR (anti-miRNA) oligonucleotides, a synthetic class of molecules typically consisting of 18-22 nucleotides, have been shown to hinder the activity of naturally occurring miRNAs by binding complementarily to their target miRNAs [198]. Within this group, LNAs represent a modified form of oligonucleotides frequently employed to inhibit oncogenic miRNAs. LNAs exhibit remarkable characteristics such as high binding affinity to complementary RNA sequences, enhanced ability to distinguish between matches and mismatches, and a low level of toxicity, rendering them among the most effective inhibitors of miRNAs [199,200]. However, the delivery of antisense oligonucleotides without an appropriate carrier faces significant challenges, primarily due to their low transfection efficiency, negative charge, hydrophilicity, and susceptibility to clearance by the reticuloendothelial system (RES) [201]. Recent advancements in receptor-mediated delivery, employing nanoscale delivery systems, offer promising prospects for achieving substantial delivery of anticancer drugs and genes. Targeted delivery systems serve to safeguard the transported genes against degradation by endonucleases and prevent their elimination by the RES. Thus, these approaches enhance the therapeutic efficacy of the delivered molecules while minimizing non-specific and undesired toxicity [200,202,203]. The utilization of anti-miR-aptamer chimers holds great potential for the efficient downregulation of miRNAs. However, there is still a lack of research on their use and effectiveness in OC [200,204].

At present, the existing literature consists solely of studies in which Vandghanooni et al. developed PEGylated poly(lactic-co-glycolic acid) NPs decorated with the AS1411 antinucleolin aptamer, loaded either with cisplatin (Ap–CIS–NPs) or anti-miR-21 (Ap-anti-miR-21-NPs), to specifically target nucleolin-overexpressing OC cells and investigate their potential in miR-21 inhibition (Figure 3D). Their investigations demonstrated that these engineered chimeras initially reduced miR-21 levels in cisplatin-resistant A2780 cells, thereby reversing drug resistance in cancer cells. Subsequently, these NPs delivered polymeric NPs carrying cisplatin and AS1411 to the cells, augmenting the cytotoxic effects of cisplatin on cancer cells. This research affirmed the crucial roles played by microRNA21 (miR-21) in tumor drug resistance, progression, and metastasis. The overexpression of miR-21 was directly associated with chemoresistance in OC, as it promoted cancer cell survival [204]. Furthermore, Vandghanooni et al. expanded their studies by loading cisplatin and locked nucleic acid (LNA) anti-miR-214 into polymeric NPs coupled with aptamer AS1411. MicroRNA214 (miR-214) exhibits overexpression in cisplatin-resistant OC cells and downregulates PTEN protein expression while activating the PI3K/Akt pathway through modulation of the 3′-UTR of the PTEN gene, facilitating cancer cell survival. Therefore, inhibiting miR-214 was shown to enhance tumor sensitivity to chemotherapeutic drugs. The study demonstrated that the tailored drug delivery systems promoted apoptosis and that targeted miR-214 inhibition using these NPs containing LNA reduced drug-resistant properties in cancer cells, potentially enhancing the efficiency of targeted drug delivery systems [200].

### 4.4. Aptamer-Decorated NPs Targeted Therapeutics

Nanotechnology holds considerable promise for advancing our understanding of pharmaceutical development, enhancing the effectiveness of therapeutic approaches, and mitigating the adverse effects associated with anti-cancer agents [205]. The utilization of NPs presents a viable strategy for encapsulating and delivering anti-cancer drugs to tumor tissues with high efficiency [183]. This approach addresses the limitations of conventional chemotherapy agents by enabling the sustained maintenance of therapeutic drug levels within cancer cells. Nano-sized lipids, polymeric NPs, and other nanoformulations offer the potential to augment the permeability of specific drugs across cell membranes, thereby improving the pharmacodynamics and pharmacokinetics of anti-cancer medications [206]. Additionally, due to the heightened permeability and retention properties exhibited by tumor cells toward NPs, the conjugation of aptamers with NPs allows for the targeted accumulation of therapeutic agents in proximity to cancer cells, enhancing overall efficacy [76]. Small ligands with specific affinity for cancer cells can be modified with NPs to enable more precise and selective transport to cancerous cells, thereby enhancing therapeutic effectiveness and reducing non-specific drug toxicity. Numerous ligands, including aptamers, have been successfully linked to NPs [76,207]. These findings underscore the versatility of NPs as vehicles for aptamers serving as effective targeting molecules [208].

Jiang et al. employed a strategy involving the encapsulation of miR-29b within cationic liposomes and subsequent tethering of these liposomes with the aptamer AS1411, known for its specificity toward nucleolin (Figure 3C). It is well established in the literature that miR-29b plays a pivotal role in inducing apoptosis, regulating cell proliferation and differentiation, and targeting genes associated with DNA methylation, cell cycle proteins, and apoptosis. The incorporation of liposomes not only enhanced drug loading capacity and circulation half-life but also endowed the chimeric aptamer/liposome/miR-29b construct with a positive charge, facilitating its interaction with negatively charged cells. This innovative approach demonstrated its effectiveness in inducing cytotoxicity in OC cancer cells A2780 [209].

Chen et al. engineered a chimera that combined a VEGF RNA aptamer with *Notch3* siRNA. They evaluated the chimera’s specific binding affinity for VEGF-overexpressing OC cells. To facilitate targeted delivery and enhance the anticancer effects, they developed innovative Au-Fe_3_O_4_ heterogeneous NPs capable of delivering the VEGF aptamer-*Notch3* siRNA chimera specifically to VEGF-positive OC cells. This delivery system effectively silenced the target *Notch3* gene and demonstrated significantly higher gene silencing efficiency when compared to the chimera alone and the lipofectamine-siRNA complex. Additionally, it improved the therapeutic efficacy of the loaded chimera. Notably, the efficient delivery of the chimera by Au-Fe_3_O_4_ NPs had the potential to reverse the MDR of OC cells to cisplatin [190].

Savla et al. reported the development and delivery of a tumor-targeted, pH-responsive chimera consisting of quantum dots (QD), a MUC1 aptamer, and doxorubicin (DOX), denoted as QD-MUC1-DOX, for the chemotherapy of OC. To achieve active cancer targeting, QD was coupled with a DNA aptamer that specifically recognizes mutated MUC1, an overexpressed protein in various cancer cells, including those in OC. DOX was linked to QD through a pH-sensitive hydrazone bond, aiming to maintain complex stability during systemic circulation and achieve drug release within the acidic intracellular environment of cancer cells. The data revealed that this hydrazone bond remains stable under neutral and slightly basic pH conditions but undergoes rapid hydrolysis in mildly acidic pH. Both confocal microscopy and in vivo imaging studies demonstrated that the developed QD-MUC1-DOX chimera exhibited increased cytotoxicity when compared to free DOX, particularly in MDR cancer cells. Additionally, it exhibited a preference for accumulating within ovarian tumors [210].

Wu et al. developed a microfluidic device based on poly(lactic-co-glycolic acid) (PLGA) nanofibers for the capture of both epithelial and mesenchymal subtypes of CTCs in OC. This device was constructed by combining a plasma-treated electrospun PLGA nanofiber chip, a polydimethylsiloxane chamber, and a glass slide. To enhance capture efficiency and facilitate effective release, bovine serum albumin (BSA) was grafted onto the surface of the PLGA nanofiber chip, serving as an adhesion resistance layer. Dual aptamers were then immobilized onto the surface of the BSA-modified PLGA nanofiber chips to selectively capture heterogeneous CTCs from simulated blood samples. Subsequently, clinical specimens obtained from OC patients were successfully detected using the dual aptamer-modified PLGA nanofiber-based microfluidic device. These findings strongly support the potential utility of dual aptamer-modified PLGA nanofiber-based microfluidic devices as a promising and reliable tool for CTC detection in clinical applications [211].

Pi et al. engineered a novel, modified nucleic acid NP chimera with distinct attributes: it featured an annexin A2 aptamer (Endo28) for precise targeting of OC cells and incorporated a GC-rich sequence to accommodate DOX. The results obtained in their study affirmed the suitability of these stable thio-DNA/2′F-RNA chimera NPs, which were equipped with the annexin A2 aptamer, as effective nanocarriers for the targeted delivery of DOX to OC cells. Notably, these DNA/RNA chimera NPs were demonstrated to retain both chemical and thermodynamic stability, rendering them well-suited for potential in vivo applications. Moreover, the annexin A2-specific NPs exhibited favorable binding profiles at an RNA concentration of 50 nM and subsequently demonstrated their capacity for precise delivery of DOX to SKOV-3 cells, resulting in significantly higher toxicity compared to targeted-scramble controls. Furthermore, when administered systemically in vivo, the Endo28 thioaptamer-harboring NPs exhibited specific targeting to OC, underscoring their potential as a promising candidate for OC therapy [212].

Ghassami et al. implemented an electrospraying technique to create NPs using poly(butylene adipate-co-butylene terephthalate) (Ecoflex^®^) loaded with docetaxel (DTX), referred to as DTX-NPs. Subsequently, HER-2 specific aptamer was added to these chimeras, forming a covalent bond and resulting in Apt-DTX-NPs. The results of their study revealed a significant increase in the in vitro cytotoxicity and cellular uptake of Apt-DTX-NPs in a cell line overexpressing HER-2, when compared to DTX-NPs and the free drug. In vivo, studies further demonstrated notable improvements in pharmacokinetic parameters, including the area under the plasma concentration-time curve, mean residence time, and elimination half-life. This was accompanied by a substantial enhancement in antitumor efficacy, which could be attributed to the targeted delivery of DTX to the tumor site and the increased cellular uptake, as substantiated by the aforementioned in vitro assessments [213].

Torabi et al. presented an innovative approach to treating cancer by using aptamers for targeted delivery of sunitinib to OC cell lines SKOV-3 and OVCAR-3. The researchers used mesoporous silica NPs due to their extensive surface area, great loading efficiency, and notable pharmacokinetic features. These NPs were coated with aptamers targeting CA125 (Mucin-16), which is a significant cancer biomarker. The results demonstrated that the designed system exhibited higher cytotoxic activity in cell lines overexpressing MUC16 (OVCAR-3) compared to MUC16 negative cell line (SKOV-3). This underscores the potential of using aptamers in the treatment of cancer with excessive expression of MUC16 (CA125) [214].

### 4.5. Conjugated Aptamers for a Multitargeting Scheme

Multivalent interactions, characterized by the simultaneous binding of two or more ligands to multiple receptors on another entity, hold significant relevance in natural biological processes such as recognition, adhesion, and signal modulation [215]. Drawing inspiration from these naturally occurring multivalent binding processes, researchers have focused their studies on the exploration of a range of multivalent aptamers designed to bind to cell surface receptors [216,217]. Shi et al. introduced the concept of multivalent aptamers and provided experimental evidence through the use of a pentavalent RNA aptamer targeting the splicing regulatory protein B52 [218]. A multivalent aptamer can be characterized as a composite comprising two or more identical or different aptamer motifs, either with or without additional structural components or functional groups [219]. A number of studies have reported that multivalent aptamers exhibit enhanced binding affinity, heightened specificity, increased resistance to nucleases, and extended circulation times compared to monomeric aptamers [215,220,221]. In contrast to chemically modified aptamers, multivalent aptamers offer advantages in terms of greater structural diversity, absence of constraints related to specific functional groups and synthetic bases, and exhibit larger molecular dimensions. These attributes enable multivalent aptamers to evade glomerular filtration, thus contributing to an extended in vivo circulation time, as elucidated [161]. Significantly, due to their distinctive structural and compositional advantages, aptamers offer the opportunity to employ a range of strategies in the development of multivalent structures as recognition elements. A prevalent approach includes hybridization, polymerization, and assembly based on nanostructures [222]. Diverse polymer-conjugated or lipid-modified aptamers have autonomously organized into multivalent aptamers designed to target cell surfaces in therapeutic applications. Furthermore, owing to advancements in nanotechnology, a plethora of nanostructures, such as NPs, nanorods, nanosheets, mesoporous structures, and DNA-based nanostructures, have been synthesized and utilized for engineering multivalent aptamers through chemical modifications [215,223]. A simple linking of identical aptamer motifs can lead to a substantial enhancement in construct avidity, primarily attributed to the presence of multiple binding sites for the target [219]. Furthermore, chemically modified aptamers can serve as fundamental units for constructing multivalent aptamers, thereby further enhancing the aptamer’s performance within complex structures [215]. Consequently, the use of multivalent aptamers holds considerable promise in expanding aptamer functions, particularly in cancer cell recognition and targeted delivery in cancer therapy.

Li et al. developed a versatile drug carrier with multivalent targeting capabilities using a self-assembled DNA nanocentipede structure. This DNA structure resembled a centipede and consisted of two main components: the “trunk” and the “legs”. The trunk was formed by the self-assembly of DNA scaffolds through a hybridization chain reaction, initiated by a trigger DNA, and the legs were biotinylated aptamers that were linked to the trunk through streptavidin-biotin affinity interactions. The elongated trunk of the DNA nanocentipede was loaded with DOX, while the legs were decorated with SMMC-7721 cell-binding aptamers (Zy1), serving as targeting elements to specifically bind to and capture target cells. The developed carrier offered several notable advantages. Compared to carriers constructed through covalent bonds, this self-assembly approach presented fewer challenges. Furthermore, the carrier demonstrated the ability to enhance binding affinity and facilitate cellular internalization without compromising selectivity due to multivalent effects. Lastly, the carrier exhibited a high drug loading capacity and efficient targeted drug delivery, resulting in improved therapeutic effectiveness and reduced drug-related side effects [224].

Zhu et al. employed a simple DNA hybridization reaction to create a novel nanocomposite, where DNA-templated fluorescent silver nanoclusters (AgNCs) clustered around a DNA-modified gold nanoparticle (AuNP). This nanocomposite, AuNP@(AS1411–AgNCs)_n_, was designed to serve dual functions: cancer cell-specific imaging and targeted therapy. The AuNP@(AS1411–AgNCs)_n_ nanocomposite exhibited robust near-infrared fluorescence emission, enhanced biostability, and was densely covered with AS1411 aptamers. Thus, the AuNP@(AS1411–AgNCs)_n_ selectively accumulated around cancer cells that displayed overexpression of nucleolin on their surfaces. Subsequently, the probe was efficiently internalized by these cancer cells, resulting in the achievement of cancer-cell-specific imaging and targeted therapy [225].

Taghavi et al. developed NPs encapsulating epirubicin through a formulation consisting of biocompatible and biodegradable PLGA modified with chitosan through a physical adsorption approach. Chitosan was utilized to modify the solubility and surface charge of PLGA, rendering it an effective drug carrier for cancer cells. To enhance anti-tumor efficacy, a targeted therapy strategy was employed using a 5TR1 DNA aptamer targeting the MUC1 receptor. In vitro, studies demonstrated that the prepared NPs, either with or without aptamers, exhibited significantly higher therapeutic effects in MCF7 cells compared to epirubicin alone. Furthermore, in BALB/c mice bearing C26 cells, the targeted NP groups displayed notable inhibition of tumor growth and exhibited a greater affinity for the tumor compared to the non-targeted NPs. Consequently, the in vivo findings suggest that non-targeted NPs may disperse away from the tumor site, leading to the release of epirubicin into the extracellular space and a reduction in drug concentration within the targeted tissue [226].

Kang et al. designed a range of linear DNA structures with varying sizes, including monomers, pentamers, nonamers, and rolling circle amplification products. These DNA structures were then combined with two distinct types of aptamers, AS1411 and MUC-1 aptamers, for a comparative assessment of the most effective DNA structures concerning intracellular uptake. The multivalent DNA structures featuring both AS1411 and MUC-1 aptamers, targeting both simultaneously, demonstrated superior intracellular uptake when compared to those containing only one type of aptamer. Notably, the internalization of DNA structures linked with aptamers increased significantly as the DNA length progressed from monomers to pentamers, nonamers, and rolling circle amplification products. Moreover, when biologically functional DNA–RNA hybrids with aptamers were administered intratumorally, multivalent nucleic-acid-based structures exhibited greater tumor accumulation than their monovalent and divalent counterparts with aptamers. Collectively, these findings underscore that both the intracellular uptake observed in vitro and the accumulation in vivo of the nucleic-acid-based structures incorporating aptamers rely significantly on multivalency and dual targeting. Furthermore, it is worth noting that all of the designed DNA–RNA hybrids exhibited minimal levels of liver toxicity and cytokine induction following intratumoral injection [227].

Alizadeh et al. prepared chitosan (CS)-coated silica NPs (SiO_2_@CS NPs) and conjugated them with epigallocatechin gallate (EGCG). The amine functional groups of chitosan in the SiO_2_@CS-EGCG NPs were electrostatically coupled with the AS1411 aptamer. The resultant NPs exhibited a core–shell structure with a spherical morphology and a size of approximately 100 nm. The release of EGCG in vitro was notably elevated under mildly acidic pH conditions, similar to the environment found in cancerous tissues. The addition of the AS1411 aptamer to the SiO_2_@CS-EGCG NPs significantly enhanced the cellular uptake of the NPs by OC SKOV-3 cells in comparison to SiO_2_@CS-EGCG NPs without the aptamer. This enhanced uptake stemmed from the specific binding of the aptamer to nucleolin present on the surface of SKOV-3 cancer cell line. Furthermore, the internalization of these NPs occurred via the macropinocytosis mechanism, thereby improving the efficiency of cellular internalization. Moreover, the developed SiO_2_@CS-EGCG–aptamer NPs showed better anticancer efficacy, which was confirmed by an increase in apoptosis and cell cycle arrest in G0/G1 phase. Additionally, real-time PCR results showed that SiO_2_@CS-EGCG–aptamer decreased the expression level ERK2 and hTERT in comparison with the SiO_2_@CS-EGCG and EGCG alone [228].

Ma et al. developed a multivalent aptamer chimera, HApt-tFNA@Dxd, by conjugating the anti-HER2 aptamer (HApt), tetrahedral framework nucleic acid (tFNA), and deruxtecan (Dxd). This chimera was designed to specifically bind to the HER2 protein found in HER2-overexpressing tumors, such as in gastric cancer or OC. HApt-tFNA@Dxd exhibited remarkable affinity to HER2-positive gastric cancer cells, both in vitro and in vivo, leading to cellular damage, microstructural alterations, and the induction of apoptosis. Upon internalization by NCI-N87 cells, HApt-tFNA@Dxd underwent lysosomal degradation, resulting in the breakdown of the HER2 protein and the chimera itself. Subsequently, Dxd was released into the cytoplasm, where it exerted its cytotoxic effects. Importantly, HApt-tFNA@Dxd demonstrated minimal adverse effects on normal tissues and vital organs while displaying heightened antitumor efficacy against HER2-positive gastric cancer when compared to the administration of free Dxd and tFNA@Dxd in NCI-N87 tumor-bearing mice [229].

## 5. Conclusions and Future Perspectives

Despite advancements in survival rates, approximately 70% of women succumb to OC within ten years of diagnosis. Considerable efforts have been devoted to identifying and detecting early-stage or minimal disease through risk assessment, preventive measures, screening techniques, and effective therapeutic interventions. Accumulating evidence highlights the substantial involvement of non-cancerous cells present in the TME and in tumorigenesis, metastasis, and resistance to chemotherapy. The recognition of the TME’s significance in the initiation and progression of OC, as well as its role in conferring resistance to anti-cancer treatments, has been increasingly acknowledged.

In recent years, aptamers have emerged as a promising class of anticancer therapeutics. Their potential applications extend beyond cancer treatment to encompass drug discovery, medical diagnostics, and drug delivery systems. Due to their exceptional targeting capabilities, aptamers have found extensive utility in the detection, diagnosis, and treatment of OC. CA125 and HE4 have gained prominence as two widely used markers in the field of OC diagnostics. These markers have received significant attention for their relevance and usefulness in various aspects of OC detection and monitoring. CA125 and HE4 have proven valuable for assessing the presence and progression of OC. Nevertheless, the current sensitivity and specificity of these markers are insufficient, highlighting the urgent requirement to investigate and establish additional diagnostic biomarkers for OC. Despite the ongoing advancement in the application of aptamers for cancer therapy, the number of aptamers that have progressed to clinical trials remains limited. Table 2 lists these aptamers, alongside their corresponding application in treating specific types of cancers.

The genetic characteristics of OC and the well-defined oncogenes provide an opportunity for the application of gene-based therapies such as RNAi. However, the use of siRNAs and miRNAs is hindered by their instability in circulation and potential toxicity in vivo due to off-target effects. In response to this limitation, a promising strategy has emerged, involving the integration of aptamers, known for their high targeting specificity, with ncRNA technology. This approach holds significant promise for attaining accurate and efficient gene targeting. Additionally, there is a growing interest in utilizing NPs conjugated with aptamers as delivery systems for siRNA or miRNA, aiming to treat both drug-resistant and drug-sensitive OC. This modification allows for the targeted delivery of ncRNAs to specific cells, tissues, or organs, facilitating their internalization with precision. Preclinical investigations have demonstrated promising outcomes, including the reduction in tumor proliferation and enhanced apoptotic effects of anticancer drugs. Nonetheless, several essential steps need to be taken before these therapeutics can have a meaningful impact at the clinical level.

## Figures and Tables

**Figure 1 cancers-15-05300-f001:**
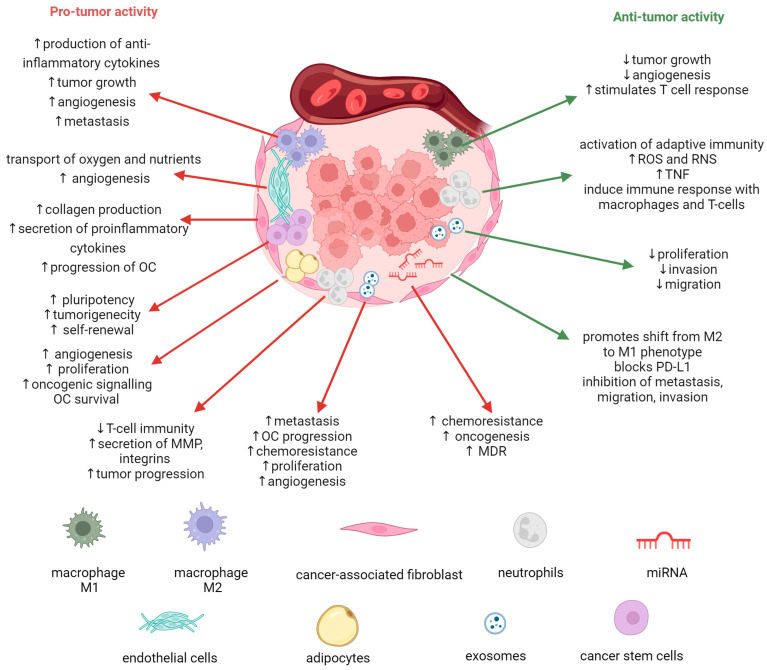
Schema of cell network in OC microenvironment. The arrows indicate pro- (in green color) and anti-tumor effects (red color).

**Figure 2 cancers-15-05300-f002:**
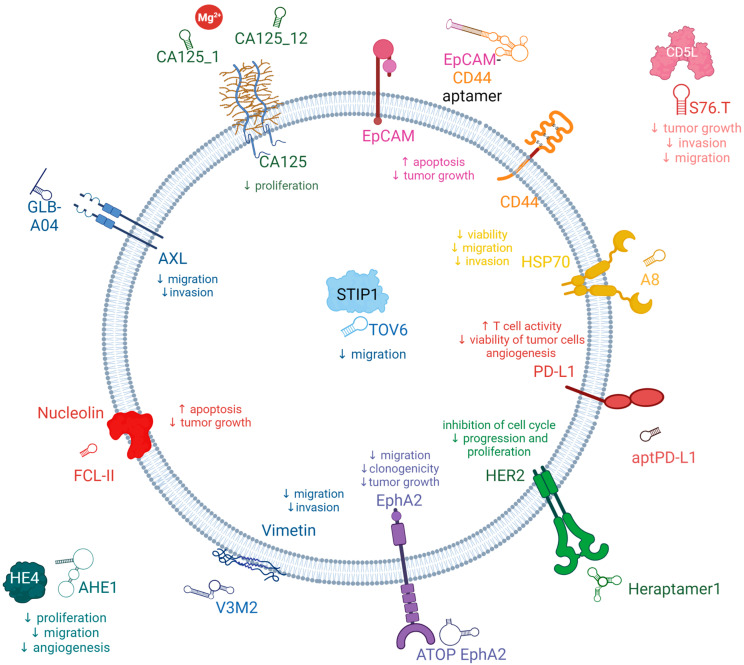
Schema of the aptamers and targets developed for the treatment of OC.

**Figure 3 cancers-15-05300-f003:**
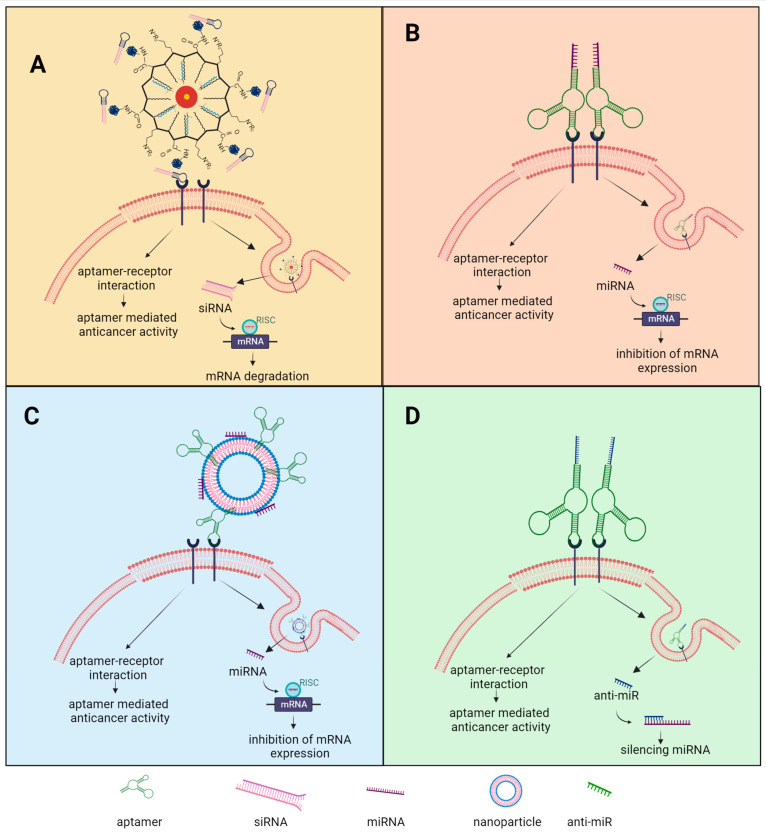
Schematic illustration of the delivery of a therapeutic agent to a target cell using: (**A**) aptamer-mediated siRNA; (**B**) aptamer-mediated miRNA; (**C**) aptamer-decorated NPs; (**D**) aptamer-anti-miR.

**Table 1 cancers-15-05300-t001:** Aptamers and their respective target molecules, along with the specific types of cancer in which they demonstrate activities.

Aptamer	Type of Cancer	Target	References
AXL-APTAMER	epithelial ovarian cancer	AXL	[71]
GLB-G25 and GLB-A04	ovarian cancer	phospho-AXL	[68]
CA125_1 and CA125_12	ovarian cancer	CA125	[81]
CA125.1 and CA125.11	ovarian cancer	CA125	[82]
rCAA-8	ovarian cancer	CA125	[54]
Apt 2.26	ovarian cancer	CA125	[83]
A1	ovarian cancer	HE4	[92]
AHE1 and AHE3	ovarian cancer	HE4	[84]
CD44-EpCAM	ovarian cancer	EpCAM and CD44	[93]
S76.T	ovarian cancer	CD5L	[106]
V3M2 and V5M2	ovarian cancer	vimentin	[107]
HA-6AS and ST-6AS	ovarian cancer	nucleolin	[118]
TOV6	ovarian cancer	STIP1	[120]
^99m^Tc-hynic-RNA aptamer	ovarian cancer	HER2	[140]
Heraptamer1 and Heraptemar2	ovarian cancer	HER2	[141]
Apt5 and Apt33	ovarian cancer	PD-L1	[57]

**Table 2 cancers-15-05300-t002:** List of clinical trials involving aptamers for therapeutic applications in the treatment of cancer.

Aptamer	Target	Type of Cancer	Phase	Status	References
AS1411	nucleolin	acute myeloid leukemia	II	completed	*NCT00512083*
AS1411	nucleolin	acute myeloid leukemia	II	terminated	*NCT01034410*
AS1411	nucleolin	advanced solid tumors	I	completed	*NCT00881244*
AS1411	nucleolin	renal cell carcinoma	II	unknown	*NCT00740441*
Sgc8	PTK7	colorectal cancer	I	unknown	*NCT03385148*
NOX-A12	CXCL12	relapsed multiple myeloma	II	completed	*NCT01521533*
NOX-A12	CXCL12	relapsed chronic lymphocytic leukemia	II	completed	*NCT01486797*
NOX-A12	CXCL12	colorectal and pancreatic cancer	I/II	completed	*NCT03168139*
NOX-A12	CXCL12	glioblastoma	I/II	active, not recruiting	*NCT04121455*
NOX-A12	CXCL12	pancreatic cancer	II	active, not yet recruiting	*NCT04901741*
EYE001	VEGF	retinal tumors	I	completed	*NCT00056199*

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
