# Peer review of "Aptamers as Potential Therapeutic Tools for Ovarian Cancer: Advancements and Challenges"

_cancers, 2023, doi:10.3390/cancers15215300_

Round 1

Reviewer 1 Report

Comments and Suggestions for Authors  the research question was very clearly addressed. The topic is original and addresses very specific question regarding the treatment of ovarian cancer which is not deeply explored in the literature The only problem of the paper is it's length, but it is un-avoidable in order to address all the aspects in this field

And all the references have been appropriately placed.

Excellent review of a very promising topic, well done!

Author Response

We thank the Reviewer for the positive comments.

Reviewer 2 Report

Comments and Suggestions for Authors

It is better to make the written form simpler, and use more pictures to explain

Comments on the Quality of English Language

It is better to make the written form simpler, and use more pictures to explain

Author Response

1) Response:

We thank the Reviewer for the positive comments and for revising our manuscript. We have shortened the paper by removing paragraphs related to cancers not related to ovarian cancer.

Reviewer 3 Report

Comments and Suggestions for Authors

In the present review, the authors discuss the use of aptamers as potential tools for the treatment of ovarian cancer (OC). The review is well-written from the language point of view and exhaustive; however it is too lengthy in its present form.  I suggest to shorten it and concentrate exclusively on OC as stated in the title.

Major:

-In section 3 “Aptamers-promising tools for cancer treatment” and relative subsections,  I would focus only on OC-related aptamers without describing in details aptamers used in different kind of cancers.  In particular, section 3.1, does not make too much sense in its current position since some papers i.e. Torabi et al., describing nanoparticles (NPs) coated with aptamers should be moved in the corresponding section where this kind of approach is described. 

- Section 3.2 and 4 could be fused, and section 4 should be reduced. It is really too long and heavy in the description of the single aptamers. In my opinion, the description should be limited to the use of these aptamers in OC models (i.e in Cerchia et al., 2012 no OC cell models are present). 

-Section 5 and subsections  should be reorganized. Section 5.4 “Aptamer-anti-miR targeted therapeutics” should become section 5.3 and current section 5.3 Aptamer-decorated NPs targeted therapeutics should be become section 5.4. 

-Section 6 should integrated in the previous sections (section 3 and subsequent sections)

Minor:

-the authors should be careful in the use of abbreviations. Ovarian cancer (OC) and nanoparticles (NPs), for example are not consistently abbreviated along the entire manuscript. Some examples below:

for ovarian cancer (OC)

Page 2 Lines 47, 49, 51, 56, 91, 96

for nanoparticles (NPs)

page 30 lines 129, 130 

-the authors should remove typos

Comments on the Quality of English Language

The review is well-written from the language point of view.

Author Response

Query 1: In the present review, the authors discuss the use of aptamers as potential tools for the treatment of ovarian cancer (OC). The review is well-written from the language point of view and exhaustive; however it is too lengthy in its present form.  I suggest to shorten it and concentrate exclusively on OC as stated in the title.

1) Response:

We thank the Reviewer for the comprehensive revisions and constructive comments. We agree with the Reviewer that the paper should focus exclusively on OC, as indicated in the title. Additionally, in accordance with the Reviewer’s suggestions, we have removed all the paragraphs no related to OC.

Query 2: In section 3 “Aptamers-promising tools for cancer treatment” and relative subsections, I would focus only on OC-related aptamers without describing in details aptamers used in different kind of cancers.  In particular, section 3.1, does not make too much sense in its current position since some papers i.e. Torabi et al., describing nanoparticles (NPs) coated with aptamers should be moved in the corresponding section where this kind of approach is described. 

2) Response:

As the Reviewer suggested, we have deleted the paragraphs describing aptamers in other types of cancers, including section 3.1. The paragraph describing the work of Torabi et al. has been relocated to a more appropriate location in section 4.4. (Please refer to lines 1302-1310).

Query 3: Section 3.2 and 4 could be fused, and section 4 should be reduced. It is really too long and heavy in the description of the single aptamers. In my opinion, the description should be limited to the use of these aptamers in OC models (i.e in Cerchia et al., 2012 no OC cell models are present). 

3) Response:

We agree with the Reviewer that section 4 is excessively long and descriptive. We have condensed and integrated it into section 3.1. Additionally, we have removed paragraphs related to different types of cancers.

Query 4: Section 5 and subsections  should be reorganized. Section 5.4 “Aptamer-anti-miR targeted therapeutics” should become section 5.3 and current section 5.3 Aptamer-decorated NPs targeted therapeutics should be become section 5.4. 

4) Response:

We thank the Reviewer for its suggestion, which will enhance the clarity and readability of the paper. We have reorganized the mentioned sections as the Reviewer suggested.

Query 5: Section 6 should integrated in the previous sections (section 3 and subsequent sections)

5) Response:

We thank the Reviewer for its suggestion. We have integrated it into section 3 as suggested. (Please refer to new heading 3.3)

Query 6: Minor:

-the authors should be careful in the use of abbreviations. Ovarian cancer (OC) and nanoparticles (NPs), for example are not consistently abbreviated along the entire manuscript. Some examples below:

for ovarian cancer (OC)

Page 2 Lines 47, 49, 51, 56, 91, 96

for nanoparticles (NPs)

page 30 lines 129, 130 

-the authors should remove typos

6) Response:

Thank the Reviewer for highlighting the inconsistent use of abbreviations in the article. To address this concern, we have thoroughly reviewed the manuscript and made the necessary revisions to ensure that abbreviations such as:

- “Ovarian cancer (OC)” in lines 38, 48, 50, 52, 57, 92, 97, 175, 179, 182, 194, 206, 253, 329, 585, 602, 614, 616, 697, 828, 849, 853, 858, 936, 970, 1211, 1212, 1216, 1219, 1220, 1224, 1227, 1229, 1333, 1342, 1415, 1417, 1431, 1438, 1447, 1500, 1667, 1689, 1706, 1840, 1841, 1850, 2003, 2029, 2124, 2467, 2496

- “nanoparticles (NPs)” in lines 87, 93, 1736, 1821, 1822, 1839, 1870, 1896, 2057, 2062, 2063, 2074, 2101, 2104, 2108, 2111, 2113, 2114, 2129, 2257, 2384, 2460, 2522

- “tumor microenvironment (TME)” in lines 169, 173, 217, 236, 253, 259, 268, 1293, 1308

are consistently abbreviated throughout the entire text. We believe that this adjustment enhances the clarity and consistency of the paper.

In the abbreviation list, we have eliminated the redundant entry “MDR – multidrug resistance.”

We have also reorganized the paper according to suggestions, limiting the number of sections to 5.

We revised the manuscript according to the reviewer’s suggestions:

  • lines 218, 331, 1876, 2371 - in vitro and in vivo have been written in italics
  • lines 727, 751, 1491, 1866 - et al. have been written in italics
  • line 1268 - the repeated “a” was removed
  • lines 467, 469, and 471 – a coma after “e.g.” has been added

Table 1 has been formatted – we have deleted lines with aptamers that demonstrate activity in cancers other than OC.